# Bayesian Data Reweighting Improves Retrieval in Knowledge-Based VQA

## Abstract

Knowledge-based Visual Question Answering (VQA) requires retrievers to incorporate external knowledge, e.g., documents, to answer questions. Existing retrievers are typically optimized with standard contrastive learning, which treats all non-positive pairs as equally informative, leading to false negative bias and difficulties in hard negative mining. To overcome these issues, we propose **Bayesian Data Reweighting (BDR)**, a probabilistic framework that assigns learnable importance weights to query-document pairs and performs Bayesian inference over these weights. We derive closed-form posterior updates under conjugate priors and develop an efficient EM algorithm for weight estimation. This approach adaptively emphasizes informative pairs without explicit hard negative mining. Experiments on two representative multimodal retrievers demonstrate consistent improvements, with BDR achieving gains of up to $8.6$ points on individual datasets and an average recall of $68.6$ across all M2KR datasets, surpassing the previous state-of-the-art. [1]

## 1 Introduction

Knowledge-based Visual Question Answering (KB-VQA) Marino et al. (2019); Schwenk et al. (2022) extends the traditional VQA task by requiring models to incorporate external knowledge sources, such as structured knowledge graphs Speer et al. (2017), unstructured textual corpora Vrandečić & Krötzsch (2014), or large-scale encyclopedic documents Mensink et al. (2023) to answer questions. These questions often involve commonsense reasoning Zellers et al. (2019), fine-grained factual knowledge Chen et al. (2023), or entity disambiguation Jian et al. (2024), which is often absent from raw visual or linguistic input. As such, KB-VQA serves as a key benchmark for evaluating a model's ability to integrate perception with world knowledge Caffagni et al. (2024); Yan & Xie (2024), and has significant implications for downstream applications in education, healthcare, and open-domain dialog systems.

Recent advances in KB-VQA have primarily focused on designing efficient multimodal retrievers, such as late interaction modules Lin et al. (2023; 2024), unified embedding architectures Jiang et al. (2025); Wei et al. (2024); Lin et al. (2025), and their combination with advanced generators Lin & Byrne (2022); Hu et al. (2023c). However, most existing retrievers Lin et al. (2023; 2024); Caffagni et al. (2025); Jiang et al. (2025) are trained with the standard InfoNCE loss Oord et al. (2018), which assumes that all non-positive samples in a batch are equally informative negatives. This assumption introduces two major limitations. First, it fails to account for *false negatives*, samples that are semantically relevant but incorrectly treated as negatives, thus pushing away potentially correct document and degrading retrieval performance Chuang et al. (2020). Second, it lacks the ability to distinguish *hard negatives* distractors that are highly similar but semantically incorrect—which can collapse the structure of the embedding space if not properly handled Wang & Liu (2021).

To mitigate the impact of false and hard negatives that exist in standard contrastive learning, we further introduce a novel **Bayesian Data Reweighting (BDR)** framework. Inspired by classical *importance sampling* Katharopoulos & Fleuret (2018), we introduce an importance weight $w_i$ for each unlabeled document $d_i$ to adjust semantic consistency among negatives. The difference between our framework and the standard contrastive learning framework is illustrated in Figure 1. We transform contrastive learning into a Bayesian reweighting problem by introducing latent importance weights

---

[1] The code is available at `https://anonymous.4open.science/r/BRCL-4403/README.md`.

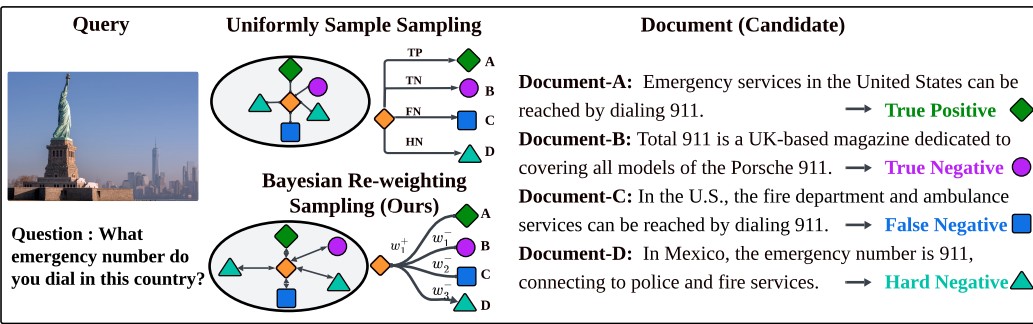

Figure 1: **Sampling strategies in KB-VQA retriever training.** Traditional methods uniformly sample negatives, treating all as equal (TN, HN, FN). Our Bayesian re-weighting instead assigns dynamic weights based on difficulty and uncertainty, mitigating false negatives and emphasizing hard negatives to refine the decision boundary.

over sample pairs. Through auxiliary variable augmentation, we achieve conditional conjugacy and tractable posteriors for these weights. The shared priors over sample-wise weights enable the model to automatically emphasize informative pairs and suppress noisy ones, without explicitly identifying hard or false negatives. Furthermore, we develop a stochastic Expectation-Maximization (EM) algorithm to jointly infer the latent variables and optimize the model parameters in a tractable and efficient manner.

We conducted experiments on two representative multimodal retrievers, Pre-FLMR Lin et al. (2024) and VLM2Vec Jiang et al. (2025), and found that applying BDR consistently improves performance. With the ViT-L backbone, Pre-FLMR with BDR achieves a $6.8$-point improvement on InfoSeek; and with ViT-G, it achieves an $8.6$-point improvement on LLaVA. For the VLM2Vec retriever, BDR brings the largest gain of 7.2 points on OKVQA when using the Phi-3.5-V-3.8B backbone. Ultimately, our best retriever, VLM2Vec with a Qwen2-VL-7B backbone, achieves an average recall of 68.6 with BDR, surpassing the previous state-of-the-art of $58.9$ on this benchmark. Equipped with our best retriever, we also achieve significant improvements in VQA accuracy across three downstream KB-VQA tasks. These results clearly demonstrate the effectiveness of the proposed BDR method for multimodal retrieval tasks. Our contributions are summarized as follows:

- We introduce **Bayesian Data Reweighting (BDR)**, a probabilistic framework that assigns importance weights to query-document pairs and performs Bayesian inference to adaptively mitigate false negative bias and facilitate hard negative mining.
- We derive closed-form posterior updates under conjugate priors and propose an efficient stochastic EM algorithm, enabling tractable and scalable optimization for scalable multi-modal retrieval tasks.
- Extensive experiments on two representative retrievers demonstrate consistent and significant improvements. In particular, BDR achieves gains of up to $8.6$ points on individual dataset and establishes a new state-of-the-art average recall of $68.6$ on the M2KR benchmark.

## 2 RELATED WORK

**Knowledge-based Visual Question Answering.** Knowledge-based Visual Question Answering (KB-VQA) extends traditional VQA by requiring external knowledge to answer questions that cannot be resolved by visual content alone Marino et al. (2019); Schwenk et al. (2022). Recent progress has shown the promise of *retrieval-augmented generation* (RAG) frameworks, where external textual resources (e.g., Wikipedia or web documents) are retrieved and fed into large multimodal language models (MLLMs) to enhance reasoning Caffagni et al. (2024); Yan & Xie (2024); Long et al. (2025). Among these, ReAuSE Long et al. (2025) tightly integrates autoregressive retrieval into the generative VQA pipeline, while Wiki-LLaVA Caffagni et al. (2024) employs a hierarchical passage retrieval strategy to select knowledge from multimodal documents. EchoSight Yan & Xie (2024) further introduces a visual retriever followed by multimodal reranking to better align visual cues with encyclopedic content. Despite these advances, KB-VQA retrieval pipelines remain brittle. often retrieving redundant or irrelevant knowledge Hao et al. (2024) or failing to capture fine-grained entities within the visual scene Jian et al. (2024), which motivates the development of a more robust and effective retriever.

**Learning to Reweight Samples in Contrastive Learning**   Contrastive learning has become a powerful paradigm for multimodal retriever optimization Misra & Maaten (2020); He et al. (2020); Chen et al. (2020); Liu et al. (2021). Existing methods typically treat paired samples as positives and all others as negatives, which introduces false negatives due to semantic overlap or label ambiguity, degrading retrieval robustness Chuang et al. (2020). Prior attempts to address this, such as hard negative mining Schroff et al. (2015), debiased contrastive loss Chuang et al. (2020), and heuristic weighting Zheng et al. (2019), often rely on fixed rules without modeling uncertainty. In contrast, our work builds on importance sampling Katharopoulos & Fleuret (2018) but differs fundamentally: (1) we introduce a Bayesian framework that infers stochastic local weights for both positives and negatives, and (2) we design a latent variable augmentation scheme enabling conjugate inference and tractable weight updates under common priors.

## 3   MULTIMODAL RETRIEVAL FRAMEWORK AND BAYESIAN REWEIGHTING

### 3.1   PRELIMINARIES

We aim to build a multimodal Retrieval-Augmented Generation (RAG) framework to enhance Knowledge-based VQA by retrieving relevant documents as external knowledge. Specifically, given an input query $x$, our framework retrieves a set of top-$k$ documents $\mathcal{D}_k(x) = \{d_1, d_2, \ldots, d_k\}$ and incorporates them as additional context to generate a target answer $y$. The overall formulation of the framework is:

$$p(y \mid x) = \underbrace{p_\theta(\mathcal{D}_k(x) \mid x)}_{\text{retriever}} \cdot \underbrace{p_\phi(y \mid x, \mathcal{D}_k(x))}_{\text{generator}}. \tag{1}$$

The Retrieval-Augmented Generation (RAG) framework consists of two components: (1) **Retriever** $p_\theta(\mathcal{D}_k(x) \mid x)$ with parameters $\theta$, which denotes the retrieval process that selects the top-$k$ most relevant documents given a query $x$ (i.e., $\mathcal{D}_k(x) = \text{TopK}(p_\theta(\cdot \mid x))$). (2) **Generator** $p_\phi(y_i \mid x, \mathcal{D}_k(x), y_{1:i-1})$ with parameters $\phi$, which generates the answer conditioned on the original query $x$ and the retrieved document set $\mathcal{D}_k(x)$. Specifically, the retriever $p_\theta(d \mid x)$ estimates the relevance of each document $d$ given query $x$, implemented by computing similarity scores between their embeddings:

$$p_\theta(d \mid x) \propto \exp\left(\mathbf{z}^\top \mathbf{q}\right), \quad \mathbf{z} = F_z(d), \quad \mathbf{q} = F_q(x), \tag{2}$$

where $F_z(\cdot)$ denoting the *document encoder* and $F_q(\cdot)$ denoting the *query encoder*, $\mathbf{z}$ and $\mathbf{q}$ are the embeddings of a document $d$ and a query $x$, respectively. We adopt Maximum Inner Product Search (MIPS) Shrivastava & Li (2014) to compute query-document similarities in sub-linear time. In multimodal RAG systems, the retriever is critical Lin & Byrne (2022), as it determines whether relevant knowledge can be retrieved. Most prior works optimize retrievers with the InfoNCE loss Oord et al. (2018), which increases similarity for positive pairs while separating negatives. Given a dataset $\mathcal{D} = \{(\mathbf{x}_i, \mathbf{d}_i)\}_{i=1}^N$, each $(\mathbf{x}_i, \mathbf{d}_i)$ is a **positive** pair, and $(\mathbf{x}_i, \mathbf{d}_j)$ with $i \neq j$ is a **negative** pair. The similarity scores are: $s_{i+} \triangleq \exp\left(\cos(\mathbf{q}_i, \mathbf{z}_i)/\tau\right), \quad s_{ik-} \triangleq \exp\left(\cos(\mathbf{q}_i, \mathbf{z}_k)/\tau\right), \quad \tau > 0$. Here we use the exponential cosine similarity, the contrastive loss is then defined as:

$$\mathcal{L}(\mathcal{D}; \boldsymbol{\theta}) = -\frac{1}{|\mathcal{D}|} \sum_{\mathbf{x}_i \in \mathcal{D}} \log\left(\mathcal{L}_{\mathbf{x}_i}\right), \text{ with } \mathcal{L}_{\mathbf{x}_i} \triangleq \frac{s_{i+}}{s_{i+} + \sum_{k=1}^K s_{ik-}}. \tag{3}$$

**Challenges**   In contrastive learning it assumes that positive and negative pairs are clean and reliable. However, in practice, negatives are randomly sampled within each batch, which may cause the False Negative problem Chuang et al. (2020) and the Hard Negative problem Robinson et al. (2020). As shown in Fig. 1, we illustrate these challenges with a multimodal query: given an image of the Statue of Liberty and the question *"What emergency number do you dial in this country?"*, the correct positive is Document A, which states that 911 is the U.S. emergency number. However, Document C ("the fire department and ambulance services can be reached by dialing 911") is semantically relevant and thus a **false negative**, while Document D, which describes emergency services in Mexico, is a **hard negative**. This example highlights the two key problems: **False negative debiasing.** Document C should not be pushed away, as it conveys the same semantic meaning as the query. **Hard negative mining.** Document D should be pushed further apart to maintain clear semantic separation between the U.S. and Mexico. The quantitative analysis of False and Hard Negatives in Appendix D.7 also shows False Negatives and Hard Negatives are prevalent in M2KR Datasets.

### 3.2 Proposed method: Bayesian Data Reweighting (BDR)

Inspired by classical *importance sampling* Katharopoulos & Fleuret (2018), we introduce an importance weight $w_i$ for each unlabeled document $d_i$ to adjust semantic consistency among negatives. We introduce local learnable weights $\{w_i^+, w_{ik}^-\}$ associated with each positive and negative pair, resulting in a weighted contrastive loss defined as:

$$\mathcal{L}^b(\mathcal{D}; \boldsymbol{\theta}) = -\frac{1}{|\mathcal{D}|} \sum_{\mathbf{x}_i \in \mathcal{D}} \log\left(\mathcal{L}_{\mathbf{x}_i}^b\right), \quad \mathcal{L}_{\mathbf{x}_i}^b \triangleq \frac{w_i^+ s_{i+}}{w_i^+ s_{i+} + \sum_{k=1}^K w_{ik}^- s_{ik-}}, \tag{4}$$

where $w_i^+$ and $w_{ik}^-$ represent the importance of the positive and negative pairs, respectively. Reasonable weights should follow these principles: (1) If a negative sample $x_{ik}^-$ is actually a false negative, $w_{ik}^-$ should be small (ideally zero) to avoid pushing apart true positives, thereby maintaining alignment. False negatives can be treated as noise, and small weights cause the gradient in equation 4 to vanish, preventing the model from learning from noisy samples. (2) If $x_{ik}^-$ is a true negative, $w_{ik}^-$ should be large to push apart hard negatives and preserve uniformity. Larger weights increase the gradient magnitude in equation 4, encouraging the model to learn decision boundaries between different semantic classes. When all weights are set to one, this loss reduces to the standard contrastive loss.

**Augmented Likelihood and Conditional Conjugacy**   The key challenge is how to assign reasonable weights that satisfy the criteria discussed above. In our framework, local weights for data pairs are inferred jointly with the global encoder parameters through Bayesian inference, without relying on a clean validation set Ren et al. (2018) or per-sample gradients Katharopoulos & Fleuret (2018). However, the weighted CL likelihood in equation 4 is generally non-conjugate under common prior choices for $w$, which makes inference intractable. To address this issue, we introduce a data-augmentation strategy that transforms the weighted CL likelihood into a conditionally conjugate form, enabling efficient posterior updates of the weights. The auxiliary variable $u_i$ follows the classical *data-augmentation* Tanner & Wong (1987) scheme as a latent variable to restore conjugacy.

First, we introduce auxiliary random variable $u_i \sim \text{Exp}(\lambda_i)$ associated with each data point, where $\lambda_i = w_i^+ s_{i+} + \sum_{k=1}^K w_{ik}^- s_{ik-}$. Using the Laplace transform identity, we have

$$\frac{1}{w_i^+ s_{i+} + \sum_{k=1}^K w_{ik}^- s_{ik-}} = \int \exp\left\{-\left(w_i^+ s_{i+} + \sum_{k=1}^K w_{ik}^- s_{ik-}\right)u_i\right\}du_i. \tag{5}$$

Given the auxiliary variable $u_i$, the conditional (unnormalized) likelihood of the sample weights takes the following exponential-family form:

$$\tilde{p}(w_i^+, w_{ik}^- \mid u_i) \propto w_i^+ s_{i+} \cdot \exp\left(-u_i w_i^+ s_{i+}\right) \cdot \prod_{k=1}^K \exp\left(-u_i w_{ik}^- s_{ik-}\right). \tag{6}$$

Here, the first term $w_i^+ s_{i+}$ comes from the numerator of the original contrastive objective, while the denominator after introducing an auxiliary variable $u_i$ yields exponential factors of the form $\exp\left[-u_i\left(w_i^+ s_{i+} + \sum_k w_{ik}^- s_{ik-}\right)\right]$. Hence, the joint likelihood in $\{w_i^+, w_{ik}^-\}$ belongs to the exponential family. We place priors on the positive/negative weights to encode different inductive biases:

$$w_i^+ \sim \text{Gamma}(a_+, b_+), \quad w_{ik}^- \sim \underbrace{\text{Gamma}(a_-, b_-)}_{\text{Continuous weighting}} \quad \text{or} \quad \underbrace{\text{Bernoulli}(p_-)}_{\text{Selective gating}} \quad \text{or} \quad \underbrace{\mathcal{N}_+(\mu, \sigma^2)}_{\text{Gaussian shrinkage}} \tag{7}$$

For the positive weights $w_i^+$, we adopt a Gamma prior because it is nonnegative, conjugate to the augmented likelihood, and its exponential special case ($a_+ = 1$) serves as a maximum-entropy prior with simple shrinkage properties. For the negative weights $w_{ik}^-$, we provide three flexible options: (i) *Gamma (continuous weighting)*, which supports values in $(0, \infty)$ and is ideal for modeling *continuous* difficulty levels of negative samples, allowing smooth and flexible weight adjustments. (ii) *Bernoulli (selective gating)*, whose outputs are restricted to $\{0, 1\}$, enabling a more aggressive keep-or-drop mechanism. (iii) *Gaussian (shrinkage around $\mu$)*, This prior reflects the assumption that most false-negative weights lie within a relatively stable interval and approximately follow a symmetrical normal-like distribution. In practice, we use a truncated Gaussian to ensure that the weights remain in the positive domain $(0, \infty)$.

Let the auxiliary variable have a Gamma prior $u_i \sim \text{Gamma}(a_u, b_u)$. Under the augmented likelihood described above, the following conditional posteriors are obtained in closed form and remain within their respective prior families.

**Theorem 3.1** (Conditional Conjugacy). *Given the augmented likelihood with auxiliary variables $u_i$, the conditional posterior distributions of the weights are:*

$$u_i \mid \{w_i^+, w_{ik}^-, \boldsymbol{\theta}\} \sim \text{Gamma}\Big(a_u, \ b_u + w_i^+ s_{i+} + \sum_k w_{ik}^- s_{ik^-}\Big), \tag{8}$$

$$w_i^+ \mid \{u_i, \boldsymbol{\theta}\} \sim \text{Gamma}(1 + a_+, \ u_i s_{i+} + b_+), \tag{9}$$

$$w_{ik}^- \mid \{u_i, \boldsymbol{\theta}\} \sim \begin{cases} \text{Gamma}(a_-, \ u_i s_{ik^-} + b_-), & \textit{(continuous weighting)} \\[2mm] \text{Bernoulli}\Big(\dfrac{p_- e^{-u_i s_{ik^-}}}{1 - p_- + p_- e^{-u_i s_{ik^-}}}\Big), & \textit{(selective gating)} \\[2mm] \mathcal{N}_+\big(\mu - \sigma^2 u_i s_{ik^-}, \ \sigma^2\big), & \textit{(Gaussian shrinkage)} \end{cases} \tag{10}$$

**Proof.** Detailed proof are provided in Appendix A.

**Efficient Inference with Stochastic Expectation Maximization**     The local weights $w_i^+$ and $w_{ik}^-$ are sample-specific latent variables whose total number scales quadratically with the dataset size, making storage and inference challenging. To address this, we propose a *stochastic Expectation-Maximization (EM)* algorithm (detailed in Appendix C) that alternates between sampling the local random variables on the fly and optimizing the global model parameters. Specifically, each EM iteration consists of: (i) a **simulation step**, where we sample the auxiliary variables $u_i$ and reweighting variables $w_i^+$ and $w_{ik}^-$ from their corresponding posteriors distribution. (ii) a **stochastic approximation step**, which updates a surrogate objective $Q_t(\boldsymbol{\theta})$ using a decaying step size schedule; and (iii) a **maximization step**, where we update $\boldsymbol{\theta}$ via stochastic gradient descent. Crucially, marginalizing out the auxiliary variables $\mathbf{u}$ from the augmented joint posterior $p(\boldsymbol{\theta}, \mathbf{u}, \{w_i^+\}, \{w_{ik}^-\} \mid \mathcal{D})$ recovers the original posterior $p(\boldsymbol{\theta}, \{w_i^+\}, \{w_{ik}^-\} \mid \mathcal{D})$, so the augmentation leaves the target inference problem unchanged.

## 4 THEORETICAL ANALYSIS

We establish two key results for the proposed Bayesian Data Reweighting (BDR): (i) consistency with supervised contrastive learning as the number of negatives grows, and (ii) a finite-sample error bound quantifying the deviation at finite $K$.

### 4.1 CONSISTENCY WITH SUPERVISED CONTRASTIVE LEARNING

**Theorem 4.1** (Consistency). *Assume $\{Z_{ik}\}_{k=1}^K$ are i.i.d. with finite second moment and $Z_{ik} \in (0, S_{\max}]$. Then, as $K \to \infty$,*

$$-\log\left(\frac{N_i}{N_i + \widehat{m}_i^{(K)}}\right) \xrightarrow{\text{p.}} -\log\left(\frac{N_i}{N_i + m_i}\right).$$

*Moreover, averaging over anchors yields* $\frac{1}{|\mathcal{D}|}\sum_i -\log\left(\frac{N_i}{N_i + \widehat{m}_i^{(K)}}\right) \xrightarrow{\text{p.}} \frac{1}{|\mathcal{D}|}\sum_i -\log\left(\frac{N_i}{N_i + m_i}\right).$

**Proof Sketch**     See Appendix B.2 for the detailed proof.

### 4.2 FINITE-SAMPLE ERROR BOUND

**Theorem 4.2** (Finite-Sample Error). *Assume $N_i \geq N_{\min} > 0$ and $Z_{ik}$ are i.i.d. sub-exponential (e.g., Gamma weights with bounded $s_{ik^-}$). Then for any $\delta \in (0, 1)$, with probability at least $1 - \delta$,*

$$\left| -\log\left(\frac{N_i}{N_i + \widehat{m}_i^{(K)}}\right) + \log\left(\frac{N_i}{N_i + m_i}\right) \right| \leq \frac{1}{N_{\min}}\left(\sqrt{\frac{2\,\text{Var}(Z_{ik})\,\log(2/\delta)}{K}} + \frac{2\,v\,\log(2/\delta)}{3K}\right),$$

*where $v$ is the sub-exponential proxy parameter of $Z_{ik}$. In particular, the deviation satisfies $|\cdot| = \mathcal{O}_{\mathbb{P}}(K^{-1/2})$, uniformly over anchors with $N_i \geq N_{\min}$.*

**Proof Sketch**     See Appendix B.3 for the detailed proof.

## 5 EXPERIMENTS

Having established the theoretical guarantees of BDR, we next evaluate its effectiveness on knowledge-intensive VQA benchmarks from two perspectives: (i) improvements in **retrieval performance**, and (ii) improvements in **answer generation performance** enabled by better retrieval.

### 5.1 TASK 1: RETRIEVAL PERFORMANCE WITH BDR

**Benchmarks and Metrics.** Our experiments are conducted on the M2KR Lin et al. (2024) benchmark, which integrates eight knowledge-intensive datasets such as OKVQA Marino et al. (2019), EVQA Mensink et al. (2023), and InfoSeek Chen et al. (2023), together with their external support documents (see Appendix D.1 for details). We evaluate BDR on two representative retrievers: PreFLMR Lin et al. (2024), built on CLIP backbones (ViT-B, ViT-L, ViT-G), and VLM2Vec Jiang et al. (2025), based on LLM backbones (Qwen2-VL Wang et al. (2024), Phi-3.5-V Abdin et al. (2024)). Performance is measured by Recall@K (R@K), which checks whether the target document is among the top-$K$ retrieved, and Pseudo Recall@K (PR@K), which checks whether any of the top-$K$ documents contain the correct answer, following prior work.

**Experimental Setup.** For the Pre-FLMR model, we trained the mapping network with a batch size of 32; for the VLM2Vec model, we trained the LoRA parameters with a LoRA rank of 4. Regarding the prior settings of BDR, empirically, the best performance was achieved with **Gamma prior**, and the parameters are $a_u = b_u = 1$, $a^+ = 2$, $b^+ = 1$, and $a^- = 5$, $b^- = 10$. Detailed results are provided in Appendix D.3. All images were resized to $224 \times 224$. Training was conducted for 2,000 steps using the Adam optimizer with a linear learning rate scheduler, starting from an initial learning rate of $2 \times 10^{-5}$. All models were trained on 4 NVIDIA A100 GPUs, and training a single VLM2Vec-based retriever typically required about 2 days.

Table 1: **Retrieval performance comparison on six knowledge-based VQA datasets from M2KR.** Results are reported in terms of Recall@5 (**R@5**) and Pseudo Recall@5 (**PR@5**). For OVEN and KVQA, we only report R@5, and for LLaVA, we only report R@1, to ensure comparability with previous baselines. **AVG** denotes the average over all metrics. Baselines are: CLIP Radford et al. (2021), ReT Caffagni et al. (2025), PreFLMR Lin et al. (2024), VLM2Vec Jiang et al. (2025). Our BDR method consistently improves over baselines across backbones and datasets.

| Retriver | Backbones | EVQA R@5 | EVQA PR@5 | OKVQA R@5 | OKVQA PR@5 | InfoSeek R@5 | InfoSeek PR@5 | OVEN R@5 | LLaVA R@1 | KVQA R@5 | Avg - |
|---|---|---|---|---|---|---|---|---|---|---|---|
| CLIP (Feature Fusion) | CLIP (ViT-B) | 21.2 | 40.5 | 9.6 | 56.0 | 19.3 | 40.4 | **59.8** | 58.0 | 22.0 | 36.3 |
| PreFLMR + InfoNCE | CLIP (ViT-B) | 55.2 | 66.6 | 25.2 | 65.6 | 25.7 | 49.4 | 45.9 | 66.9 | 29.7 | 47.8 |
| PreFLMR + BDR (Ours) | CLIP (ViT-B) | **55.5** | **66.8** | **29.2** | **68.2** | **26.3** | **49.8** | 49.8 | **69.7** | **32.2** | **49.7** |
| Δ | | +0.3 | +0.2 | +4.0 | +2.6 | +0.6 | +0.4 | +3.9 | +2.8 | +2.5 | +1.9 |
| CLIP (Feature Fusion) | CLIP (ViT-L) | 35.6 | 52.6 | 12.1 | 59.4 | 38.2 | 54.7 | **76.0** | 63.6 | **47.5** | 48.9 |
| PreFLMR + InfoNCE | CLIP (ViT-L) | 60.7 | 71.0 | 27.8 | 67.5 | 36.0 | 56.4 | 59.8 | 72.0 | 42.9 | 54.9 |
| PreFLMR + BDR (Ours) | CLIP (ViT-L) | **60.9** | **71.4** | **31.6** | **70.5** | **42.8** | **59.2** | 65.8 | **74.8** | 46.6 | **58.2** |
| Δ | | +0.2 | +0.4 | +3.8 | +3.0 | +6.8 | +2.8 | +6.0 | +2.8 | +3.7 | +3.3 |
| ReT | OpenCLIP (ViT-G) | 48.6 | 60.2 | 19.0 | 63.8 | 52.0 | 62.5 | 84.0 | 79.2 | 60.6 | 58.9 |
| PreFLMR + InfoNCE | OpenCLIP (ViT-G) | 62.0 | 72.0 | 30.2 | 67.4 | 39.2 | 57.7 | 64.3 | 72.6 | 41.9 | 56.4 |
| PreFLMR + BDR (Ours) | OpenCLIP (ViT-G) | **62.1** | **72.1** | **32.5** | **67.8** | 43.8 | 59.1 | 67.6 | **81.2** | 49.8 | **59.6** |
| Δ | | +0.1 | +0.1 | +2.3 | +0.4 | +4.6 | +1.4 | +3.3 | +8.6 | +7.9 | +3.2 |
| VLM2Vec (Zero-shot) | Qwen-2-VL-2B | 10.9 | 29.3 | 9.4 | 32.0 | 10.2 | 20.6 | 41.0 | 51.0 | 28.9 | 25.9 |
| VLM2Vec + InfoNCE | Qwen-2-VL-2B | 50.4 | 63.9 | 24.8 | 58.7 | 58.5 | 53.7 | 75.6 | 84.2 | 51.0 | 57.9 |
| VLM2Vec + BDR (Ours) | Qwen-2-VL-2B | **51.2** | **64.2** | **26.6** | **59.7** | **60.5** | **56.7** | **78.3** | **88.7** | **55.6** | **60.2** |
| Δ | | +0.8 | +0.3 | +1.8 | +1.0 | +2.0 | +3.0 | +2.7 | +4.5 | +4.6 | +2.3 |
| VLM2Vec (Zero-shot) | Phi-3.5-V-3.8B | 18.8 | 41.2 | 13.3 | 58.5 | 12.0 | 25.8 | 51.3 | 71.5 | 34.9 | 36.3 |
| VLM2Vec + InfoNCE | Phi-3.5-V-3.8B | 45.1 | 60.3 | 35.1 | 65.3 | 40.8 | 44.3 | 71.5 | 91.4 | 52.6 | 56.3 |
| VLM2Vec + BDR (Ours) | Phi-3.5-V-3.8B | **49.2** | **62.2** | **42.3** | **69.1** | 43.8 | **47.5** | **74.7** | **91.6** | **57.7** | **59.8** |
| Δ | | +4.1 | +1.9 | +7.2 | +3.8 | +3.0 | +3.2 | +3.2 | +0.2 | +5.1 | +3.5 |
| VLM2Vec (Zero-shot) | Qwen2-VL-7B | 18.2 | 42.8 | 13.4 | 58.0 | 14.3 | 29.8 | 63.8 | 50.1 | 42.2 | 36.9 |
| VLM2Vec + InfoNCE | Qwen2-VL-7B | 62.0 | 70.8 | 41.4 | 68.7 | 64.6 | 58.4 | 80.9 | 90.0 | 63.2 | 66.6 |
| VLM2Vec + BDR (Ours) | Qwen2-VL-7B | **64.3** | **73.1** | **43.5** | **69.9** | **66.7** | **60.5** | **83.4** | 91.0 | **65.3** | **68.6** |
| Δ | | +2.3 | +2.3 | +2.1 | +1.2 | +2.1 | +2.1 | +2.5 | +1.0 | +2.1 | +2.0 |

Table 2: **Answer generation performance comparison on InfoSeek and EVQA.** We report VQA Accuracy, Exact Match (EM), BLEU-1, and BERT Matching (BEM). The Oracle Retriever retrieves all ground-truth documents. Our BDR retriever consistently outperforms PreFLMR Lin et al. (2024) and ReT Caffagni et al. (2025), approaching the Oracle upper bound.

| Generator (Frozen) | Retriever | R@5 | InfoSeek VQA_Acc | EM | BLEU_1 | R@5 | EVQA VQA_Acc | EM | BLEU_1 | BEM |
|---|---|---|---|---|---|---|---|---|---|---|
| LLaVA-1.6-13B | ✗ | - | 5.4 | 5.3 | 11.9 | - | 2.7 | 7.4 | 8.9 | 69.8 |
| LLaVA-1.6-13B | PreFLMR | 39.2 | 12.9 | 12.4 | 21.2 | 62.0 | 8.7 | 20.5 | 26.2 | 74.3 |
| LLaVA-1.6-13B | ReT | 52.0 | 17.3 | 17.2 | 28.9 | 48.6 | 6.5 | 14.6 | 19.2 | 73.2 |
| LLaVA-1.6-13B | VLM2Vec-BDR (Ours) | 66.7 | **20.8** | **20.9** | **34.0** | **64.3** | **9.1** | **21.2** | **26.9** | **77.2** |
| LLaVA-1.6-13B | Oracle Retriever | - | 37.5 | 39.5 | 56.4 | - | 16.1 | 37.7 | 46.1 | 86.7 |
| Qwen2.5-VL-7B | ✗ | - | 14.4 | 14.5 | 25.2 | - | 4.6 | 12.0 | 14.3 | 65.2 |
| Qwen2.5-VL-7B | PreFLMR | 39.2 | 21.5 | 16.1 | 24.1 | 62.0 | 11.5 | 29.0 | 34.7 | 68.3 |
| Qwen2.5-VL-7B | ReT | 52.0 | 25.9 | 21.5 | 32.2 | 48.6 | 10.8 | 22.7 | 28.1 | 67.9 |
| Qwen2.5-VL-7B | VLM2Vec-BDR (Ours) | 66.7 | **32.1** | **27.5** | **41.3** | **64.3** | **14.4** | **30.1** | **37.1** | **71.2** |
| Qwen2.5-VL-7B | Oracle Retriever | - | 46.2 | 41.3 | 61.9 | - | 23.3 | 46.8 | 57.8 | 89.1 |

**Main Results.** For CLIP-based architectures, **applying our BDR method consistently improves the performance of Pre-FLMR**, as shown in Table 1. Specifically, with the ViT-B backbone, our BDR brings a +4.0 gain in on OKVQA; with ViT-L, it yields a +6.8 Recall@5 gain on InfoSeek; and with ViT-G, it achieves the largest improvement of +8.6 on LLaVA. For the LLM-based retriever VLM2Vec, **BDR also delivers notable improvements across different LLM backbones**: Qwen2-VL-2B achieves +4.6 on KVQA, Qwen2-VL-7B achieves +2.3 on EVQA, and Phi-3.5-V-3.8B achieves the largest gain of +7.2 on OKVQA. These results demonstrate that BDR consistently enhances retrieval performance across diverse architectures. Moreover, **our best retriever with BDR establishes a new state of the art on the M2KR benchmark**, where VLM2Vec + BDR (Qwen2-VL-7B) achieves an average recall of 68.6, surpassing the previous best result by PreFLMR Lin et al. (2024) (56.4) and ReT Caffagni et al. (2025) (58.9). These experiments clearly validate the effectiveness of the proposed BDR method for multimodal retrieval tasks.

## 5.2 Task 2: Answer Generation Performance with BDR

**Experimental Setup.** To evaluate the performance of different retrievers on downstream VQA tasks, we conduct experiments on three benchmarks: InfoSeek, EVQA, and OKVQA. We use LLaVA-1.6-13B Liu et al. (2023a) and Qwen2.5-VL-7B Team (2025) as generators in combination with different retrievers, and adopt three evaluation metrics like VQA Accuracy, Exact Match (EM), BLEU-1 and BERT Matching (BEM) to measure how well the generated answers align with the ground-truth.

**Results.** On the answer generation task in the InfoSeek and EVQA datasets, **the generator also achieves significant improvements due to our BDR retriever**. Table 2 reports the results on the InfoSeek and EVQA benchmarks. Without retrievers, both models perform poorly across all metrics, highlighting **the necessity of external knowledge retrieval for knowledge-intensive VQA**. Incorporating PreFLMR or ReT yields consistent improvements, but their gains remain limited. In contrast, **our proposed BDR retriever achieves substantial performance boosts** on both datasets. Specifically, with LLaVA-1.6-13B as the generator, our BDR yields a +3.5 gain ($17.3 \rightarrow 20.8$) in VQA Accuracy on InfoSeek, with Qwen2.5-VL-7B as the generator, it improves VQA Accuracy by +6.2 ($25.9 \rightarrow 32.1$) on InfoSeek. Moreover, on the EVQA benchmark, our BDR retriever achieves the best BEM scores among all non-oracle retrievers (e.g., **77.2** vs. 74.3/73.2 with LLaVA-1.6-13B and **71.2** vs. 68.3/67.9 with Qwen2.5-VL-7B), demonstrating substantially stronger knowledge grounding. Compare with the Oracle Retriever, **our method significantly narrows the gap with the oracle retriever**, demonstrating the effectiveness of BDR retriver in enhancing knowledge-intensive VQA.

On the answer generation task in the OKVQA dataset, equipped with our best BDR retriever, **a medium-scale LLM generator outperforms both fine-tuned generators and large LLM-based generators** such as GPT-4V, as shown in Table 3. Traditional methods combining fine-tuned generators with DPR retrievers achieve moderate performance (58 VQA-Acc), while large proprietary LLMs such as Flamingo-80B and GPT-4V reach 64.3 without retrievers but incur prohibitive costs. In contrast, combining medium-scale generators (Qwen2.5-VL-7B and LLaVA-1.6-13B) with our proposed **VLM2Vec-BDR** retriever yields substantial gains, improving VQA-Acc by +4.3 points

Table 3: **Answer generation performance comparison on OKVQA.** We report and VQA Accuracy (VQA-Acc). Baselines are divided into two categories: (i) fine-tuned generators and (ii) large language models used without fine-tuning. With our BDR retriever, a medium-scale LLM (Qwen2.5-VL-7B or LLaVA-1.6-13B) achieves superior performance, surpassing both fine-tuned generators and large LLM-based generators such as GPT-4V.

| Generator | Generator FT | Retriever | PR@5 | Knowledge Source | VQA-Acc |
|---|---|---|---|---|---|
| TRiG Gui et al. (2021) | ✓ | DPR | - | Wikipedia | 50.5 |
| RA-VQA Lin & Byrne (2022) | ✓ | DPR | - | Google Search | 51.2 |
| KAT Gui et al. (2021) | ✓ | ✗ | - | Wikipedia + GPT-3 | 54.4 |
| TWO Si et al. (2023) | ✓ | DPR | - | VQAv2 + Wikipedia | 56.7 |
| REVIVE Lin et al. (2022) | ✓ | ✗ | - | Wikipedia + GPT-3 | 58.0 |
| Flamingo-80B Alayrac et al. (2022) | ✗ | ✗ | - | Chinchilla | 57.8 |
| PromptCap-175B Hu et al. (2023b) | ✗ | ✗ | - | GPT3 | 60.4 |
| Prophet-175B Shao et al. (2023) | ✗ | ✗ | - | GPT3 | 61.1 |
| GPT4-V Achiam et al. (2023) | ✗ | ✗ | - | - | 64.3 |
| Qwen2.5-VL-7B | ✗ | ✗ | - | - | 62.4 |
| Qwen2.5-VL-7B | ✗ | PreFLMR | 67.4 | Google Search | 64.3 |
| Qwen2.5-VL-7B | ✗ | VLM2Vec-BDR (Ours) | 69.9 | Google Search | **66.7** |
| LLaVA-1.6-13B | ✗ | ✗ | - | - | 61.9 |
| LLaVA-1.6-13B | ✗ | PreFLMR | 67.4 | Google Search | 65.5 |
| LLaVA-1.6-13B | ✗ | VLM2Vec-BDR (Ours) | 69.9 | Google Search | **68.0** |

$(62.4 \rightarrow 66.7)$ and $+6.1$ points $(61.9 \rightarrow 68.0)$, respectively. These results surpass PreFLMR baselines and even outperform some ultra-large LLMs, demonstrating that our retriever provides an efficient and effective alternative to scaling model size for knowledge-intensive VQA.

### 5.3 WHY THE BAYESIAN DATA REWEIGHTING FRAMEWORK WORKS?

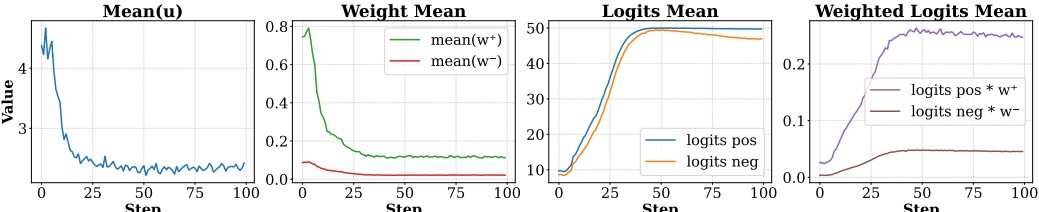

Figure 2: **Training dynamics of weighted contrastive learning.** From left to right: (1) Mean of $u$; (2) Mean of weights $w^+$ and $w^-$; (3) Mean of positive and negative logits; (4) Mean of weighted logits. These curves show how learned weights shape the contrastive signal during training.

Our experimental results demonstrate that BDR effectively addresses both **false negatives** and **hard negatives** in contrastive learning, showing in Figure 2. Specifically, the auxiliary variable $u$ rapidly decreases and stabilizes in the early training phase, providing global regularization and ensuring convergence for subsequent weight updates. The positive and negative weights, $w^+$ and $w^-$, then exhibit distinct dynamics: false negatives are suppressed with near-zero $w^-$, minimizing their interference, while truly informative hard negatives retain moderate weights, allowing the model to learn discriminative features. Most importantly, when applying these weights, **the margin between weighted positive and negative logits is significantly enlarged**, even when the raw logits are close, thus maintaining **a stable and clear decision boundary**.

### 5.4 ANALYSIS OF THE IMPACT OF TOP-K RETRIEVAL

In general, **increasing the number of retrieved documents improves answer generation, but the gain saturates quickly**. As shown in Figure 3, Qwen2.5-VL-7B reaches its best performance with Top-5 document retrieval on InfoSeek and OKVQA, while results on EVQA remain largely unaffected. This underscores that retriever performance is more important than document quantity.

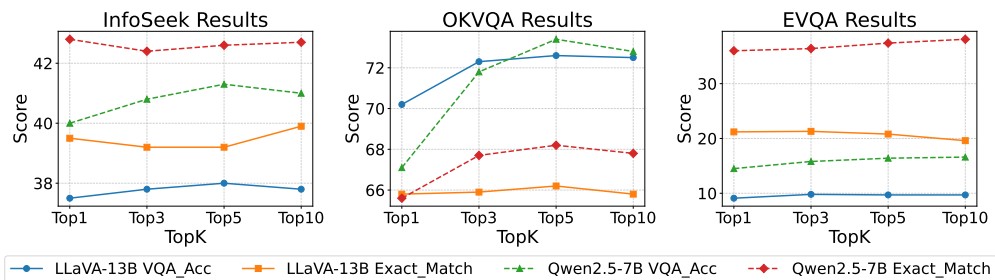

Figure 3: **Impact of retrieved Top-K documents on VQA performance.** We report VQA Accuracy (VQA-Acc) and Exact Match (EM) on InfoSeek (left) and OKVQA (right).

## 5.5 ANALYSIS OF MODEL EFFICIENCY

**Training Efficiency.** BDR adds only lightweight computations to InfoNCE. Each SAEM iteration includes: (i) an *E-step* that samples $u_i, w_i^+, w_{ik}^-$ with element-wise closed-form updates costing $O(BK)$; (ii) a *stochastic-approximation step* with negligible $O(1)$ cost; and (iii) an *M-step* identical to the standard InfoNCE encoder forward–backward pass. Thus, the total complexity per iteration is $O(\text{Encoder}) + O(BK)$, and since $O(\text{Encoder}) \gg O(BK)$ in modern vision and LLM-based retrievers, the overhead introduced by SAEM is negligible. We further validate this by training the retriver on the OKVQA dataset for one epoch and measuring total runtime. As shown in Table 4, BDR exhibits runtime and memory usage nearly identical to InfoNCE, confirming that the SAEM updates do not affect overall training speed.

Table 4: **Model training efficiency comparison.** We report training steps and and total time for VLM2Vec with InfoNCE and BDR. Note that increasing the batch size does not significantly increase memory usage, thanks to the GradCache mechanism Jiang et al. (2025) in VLM2Vec which decouples encoder backpropagation from the contrastive loss.

| Model | Backbone | Steps | GPU (GB) | Time (h) |
|---|---|---|---|---|
| **Batch Size = 32** | | | | |
| InfoNCE | Phi-3.5-V | 521 | 19.9 | 3.17 |
| BDR (Ours) | Phi-3.5-V | 521 | 20.0 | 3.22 |
| InfoNCE | Qwen2-VL-7B | 521 | 32.8 | 2.01 |
| BDR (Ours) | Qwen2-VL-7B | 521 | 32.9 | 2.05 |
| **Batch Size = 128** | | | | |
| InfoNCE | Phi-3.5-V | 131 | 19.8 | 3.16 |
| BDR (Ours) | Phi-3.5-V | 131 | 20.3 | 3.18 |
| InfoNCE | Qwen2-VL-7B | 131 | 32.8 | 2.00 |
| BDR (Ours) | Qwen2-VL-7B | 131 | 32.9 | 2.02 |

**Inference Efficiency.** Our lightweight VLM2Vec-BDR model achieves a strong balance between speed and accuracy. As the first to apply an LLM-based retriever to KB-VQA, we also evaluate its inference efficiency. The retrieval statistics on EVQA are shown in Table 5. Using the Qwen2-VL-2B backbone, VLM2Vec-BDR reduces total retrieval time to **1285s**, nearly half the cost of traditional P-FLMR retrievers, while maintaining a high average recall of **60.2**. This demonstrates that VLM2Vec-BDR offers a practical and efficient solution for knowledge-intensive VQA.

Table 5: **Retrieval efficiency comparison.** We report query encoding, passage encoding, and retrieval time for different retrievers. VLM2Vec-BDR (Qwen2-VL-2B) achieves the best trade-off between speed and accuracy.

| Model | Backbone | Qry (s) | Psg (s) | Ret (s) | Total (s) | Recall |
|---|---|---|---|---|---|---|
| P-FLMR | CLIP-B | 213 | 2174 | 62 | 2449 | 47.8 |
| P-FLMR | CLIP-L | 256 | 2163 | 61 | 2479 | 54.9 |
| P-FLMR | CLIP-G | 442 | 2174 | 61 | 2677 | 56.4 |
| BDR (Ours) | Qwen-7B | 172 | 2903 | 17 | 3091 | 68.6 |
| BDR (Ours) | Phi-3.5V | 301 | 2689 | 14 | 3004 | 59.8 |
| BDR (Ours) | Qwen-2B | 69 | 1209 | 8 | **1285** | **60.2** |

## 6 CONCLUSION

In this work, we introduced **Bayesian Data Reweighting (BDR)**, a principled framework that addresses the limitations of standard contrastive learning in multimodal retrieval. By inferring stochastic importance weights for positives and negatives, BDR naturally mitigates false negatives and emphasizes hard negatives through a Bayesian inference mechanism. Extensive experiments on the M2KR benchmark demonstrate consistent gains across both CLIP- and LLM-based retrievers. Furthermore, integrating BDR with VLM2Vec significantly boosts knowledge-intensive VQA performance, surpassing strong baselines. These results highlight BDR as a robust and efficient solution for advancing multimodal retrieval and knowledge-based answering.

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

## A    DETAILED PROOFS FOR CONDITIONAL CONJUGACY

**Parameterization and Notation.**    We use the *Gamma(shape, rate)* parameterization throughout. For each anchor $i$, let

$$s_{i+} \triangleq \exp\big(\cos(\mathbf{q}_i, \mathbf{z}_i)/\tau\big), \quad s_{ik-} \triangleq \exp\big(\cos(\mathbf{q}_i, \mathbf{z}_k)/\tau\big), \;\; \tau > 0 \tag{1}$$

which are constants with respect to the weights $\{w_i^+, w_{ik}^-\}$ and the auxiliary variable $u_i$ when $\boldsymbol{\theta}$ is fixed. The per-sample weighted-CL likelihood contribution can be written as

$$\mathcal{L}_{\mathbf{x}_i}^r = \frac{w_i^+ s_{i+}}{w_i^+ s_{i+} + \sum_{k=1}^K w_{ik}^- s_{ik-}} \quad \implies \quad \log \mathcal{L}_{\mathbf{x}_i}^r = \log\big(w_i^+ s_{i+}\big) - \log\Big(w_i^+ s_{i+} + \sum_{k=1}^K w_{ik}^- s_{ik-}\Big) \tag{2}$$

Define the shorthand $\lambda_i \triangleq w_i^+ s_{i+} + \sum_{k=1}^K w_{ik}^- s_{ik-}$.

**Lemma A.1** (Laplace augmentation identity). *For any $\lambda > 0$, $\frac{1}{\lambda} = \int_0^\infty e^{-\lambda u}\, du$.*

Applying Lemma A.1 to $1/\lambda_i$ and exponentiating $\log(w_i^+ s_{i+})$ yields the following *unnormalized* augmented likelihood kernel for a single $i$:

$$\tilde{p}(w_i^+, \{w_{ik}^-\}, u_i \mid \boldsymbol{\theta}) \;\propto\; \big(w_i^+ s_{i+}\big) \exp\Big(-u_i\, w_i^+ s_{i+}\Big) \prod_{k=1}^K \exp\Big(-u_i\, w_{ik}^- s_{ik-}\Big), \quad u_i \geq 0 \tag{3}$$

We further place priors

$$u_i \sim \text{Gamma}(a_u, b_u), \quad w_i^+ \sim \text{Gamma}(a_+, b_+), \quad w_{ik}^- \sim \begin{cases} \text{Gamma}(a_-, b_-), & \text{(continuous weighting)} \\ \text{Bernoulli}(p_-), & \text{(selective gating)} \\ \mathcal{N}(\mu, \sigma^2), & \text{(Gaussian shrinkage)} \end{cases}$$

with all priors independent across $i, k$. Conditioned on $\boldsymbol{\theta}$ (hence $s_{i+}, s_{ik-}$ are fixed), the joint posterior factorizes conveniently, enabling closed-form conditionals.

### A.1    CONDITIONAL OF $u_i$

From equation 3, the *only* dependence on $u_i$ is via $\exp\big(-u_i \lambda_i\big)$. Combining with $u_i \sim \text{Gamma}(a_u, b_u)$ gives

$$p(u_i \mid w_i^+, \{w_{ik}^-\}, \boldsymbol{\theta}) \;\propto\; u_i^{a_u-1} \exp\Big(-(b_u+\lambda_i)\, u_i\Big) \;\Rightarrow\; u_i \mid w_i^+, \{w_{ik}^-\}, \boldsymbol{\theta} \sim \text{Gamma}\big(a_u,\, b_u+\lambda_i\big), \tag{4}$$

i.e.,

$$u_i \mid \{w_i^+, w_{ik}^-, \boldsymbol{\theta}\} \sim \text{Gamma}\bigg(a_u,\, b_u + w_i^+ s_{i+} + \sum_{k=1}^K w_{ik}^- s_{ik-}\bigg). \tag{5}$$

### A.2    CONDITIONAL OF $w_i^+$ UNDER A GAMMA PRIOR

Using equation 3 and the prior $w_i^+ \sim \text{Gamma}(a_+, b_+)$,

$$p(w_i^+ \mid u_i, \boldsymbol{\theta}) \;\propto\; \underbrace{(w_i^+)^{a_+-1} e^{-b_+ w_i^+}}_{\text{Gamma prior}} \cdot \underbrace{(w_i^+ s_{i+})\, e^{-u_i s_{i+} w_i^+}}_{\text{aug. likelihood in } w_i^+} \;\propto\; (w_i^+)^{(a_+-1)+1} \exp\Big(-(b_+ + u_i s_{i+})\, w_i^+\Big),$$

which is the kernel of a Gamma with updated shape/rate. Hence

$$w_i^+ \mid u_i, \boldsymbol{\theta} \sim \text{Gamma}\Big(a_+ + 1,\, b_+ + u_i s_{i+}\Big). \tag{6}$$

## A.3 CONDITIONAL OF $w_{ik}^-$: THREE PRIOR CHOICES

**(a) Gamma prior (continuous weighting).** From equation 3, for a fixed $k$ the $w_{ik}^-$-dependent term is $\exp(-u_i s_{ik-} w_{ik}^-)$. Multiplying by the prior $w_{ik}^- \sim \mathrm{Gamma}(a_-, b_-)$ gives

$$p(w_{ik}^- \mid u_i, \boldsymbol{\theta}) \propto (w_{ik}^-)^{a_- - 1} \exp\Big( -(b_- + u_i s_{ik-}) \, w_{ik}^- \Big),$$

which is Gamma with shape $a_-$ and rate $b_- + u_i s_{ik-}$. Therefore

$$w_{ik}^- \mid u_i, \boldsymbol{\theta} \sim \mathrm{Gamma}\Big(a_-, \, b_- + u_i s_{ik-}\Big). \tag{7}$$

**(b) Bernoulli prior (selective gating).** Now $w_{ik}^- \in \{0, 1\}$ with prior $\Pr(w_{ik}^- = 1) = p_-$. The augmented likelihood factor for $w_{ik}^-$ is $\exp(-u_i s_{ik-} w_{ik}^-)$. Thus, up to a shared normalizer:

$$\Pr(w_{ik}^- = 1 \mid u_i, \boldsymbol{\theta}) \propto p_- \, e^{-u_i s_{ik-}}, \qquad \Pr(w_{ik}^- = 0 \mid u_i, \boldsymbol{\theta}) \propto 1 - p_-.$$

Hence the posterior success probability is

$$\Pr(w_{ik}^- = 1 \mid u_i, \boldsymbol{\theta}) = \frac{p_- e^{-u_i s_{ik-}}}{(1 - p_-) + p_- e^{-u_i s_{ik-}}} \tag{8}$$

$$\implies \quad w_{ik}^- \mid u_i, \boldsymbol{\theta} \sim \mathrm{Bernoulli}\left( \frac{p_- e^{-u_i s_{ik-}}}{1 - p_- + p_- e^{-u_i s_{ik-}}} \right). \tag{9}$$

**(c) Gaussian prior (shrinkage around $\mu$).** Let $w_{ik}^- \sim \mathcal{N}(\mu, \sigma^2)$. The augmented likelihood in $w_{ik}^-$ contributes $\exp(-u_i s_{ik-} w_{ik}^-)$, which is linear in $w_{ik}^-$. Completing the square:

$$\underbrace{\exp\left[ -\frac{(w_{ik}^- - \mu)^2}{2\sigma^2} \right]}_{\text{Gaussian prior}} \cdot \underbrace{\exp\left( -u_i s_{ik-} w_{ik}^- \right)}_{\text{likelihood factor}} \propto \exp\left\{ -\frac{1}{2\sigma^2} \Big[ w_{ik}^{-2} - 2\mu w_{ik}^- \Big] - u_i s_{ik-} w_{ik}^- \right\}$$

$$= \exp\left\{ -\frac{1}{2\sigma^2} \Big[ w_{ik}^{-2} - 2(\mu - \sigma^2 u_i s_{ik-}) \, w_{ik}^- \Big] \right\} \propto \exp\left[ -\frac{(w_{ik}^- - (\mu - \sigma^2 u_i s_{ik-}))^2}{2\sigma^2} \right],$$

which is the kernel of a Normal with the *same* variance and a shifted mean. Therefore

$$w_{ik}^- \mid u_i, \boldsymbol{\theta} \sim \mathcal{N}\Big( \mu - \sigma^2 u_i s_{ik-}, \, \sigma^2 \Big). \tag{10}$$

## A.4 SUMMARY OF THE CONDITIONALS

Collecting equation 5, equation 6, equation 7, equation 9, and equation 10, we obtain the conditional posteriors stated in Theorem 3.1.

*Remark* A.2 (Support and mild regularity). The augmentation identity in Lemma A.1 requires $\lambda_i > 0$. This is automatically satisfied when $s_{i+}, s_{ik-} \geq 0$ and $w_i^+, w_{ik}^- \geq 0$ (Gamma/Bernoulli cases). For the Gaussian case where $w_{ik}^-$ has full real support, the conditional updates equation 10 remain valid, and the conditional of $u_i$ in equation 5 is proper as long as $b_u + \lambda_i > 0$. In practice one may (i) pick $b_u > 0$ sufficiently large, (ii) clip or reparameterize $w_{ik}^-$ (e.g., via $\mathrm{softplus}$) if needed, or (iii) work with $\tilde{s} \geq 0$ (which holds by construction).

# B DETAILED PROOFS OF CONSISTENCY AND ERROR BOUND

We provide formal guarantees for the proposed Bayesian Data Reweighting (BDR) objective

$$\mathcal{L}^r(\mathcal{D}; \boldsymbol{\theta}) = -\frac{1}{|\mathcal{D}|} \sum_{\mathbf{x}_i \in \mathcal{D}} \log \left( \frac{w_i^+ s_{i+}}{w_i^+ s_{i+} + \sum_{k=1}^K w_{ik}^- s_{ik-}} \right),$$

where $w_i^+$ and $w_{ik}^-$ are sample-specific latent weights inferred under our Bayesian augmentation scheme, and $s_{i+}, s_{ik-} \in (0, S_{\max}]$ denote positive and negative similarities for anchor $i$. We study the asymptotics as the number of negatives per anchor $K \to \infty$ and derive a finite-sample concentration bound in $K$.

## B.1 Assumptions and Notation

For each anchor $i$, let $\{(w_{ik}^-, s_{ik^-})\}_{k=1}^K$ be i.i.d. conditional on the anchor and global parameters $\boldsymbol{\theta}$, with the following assumptions.

**A1 (Bounded similarity)** There exists $S_{\max} < \infty$ such that $s_{i+}, s_{ik^-} \in (0, S_{\max}]$ almost surely.

**A2 (i.i.d. negatives)** For fixed $i$, $\{(w_{ik}^-, s_{ik^-})\}_{k=1}^K$ are i.i.d. draws from a stationary data-generating process conditional on the anchor $i$ and current $\boldsymbol{\theta}$.

**A3 (Moment bounded weights)** There exists $W_1, W_2 < \infty$ such that $\mathbb{E}[w_{ik}^-] \le W_1$ and $\mathbb{E}[(w_{ik}^-)^2] \le W_2$ for all $i, k$. Moreover $w_{ik}^- \ge 0$ almost surely.

**A4 (Positive-part stability)** $w_i^+ \in (0, W_+^{\max}]$ for some finite $W_+^{\max}$ almost surely (or in probability), and independent of $\{(w_{ik}^-, s_{ik^-})\}_{k=1}^K$ conditional on $(\mathbf{x}_i, \boldsymbol{\theta})$.[2]

**A5 (True-negative target)** There exists a target (supervised) true-negative distribution $\mathcal{P}_{\mathrm{TN}}$ such that $\mu_i^{\mathrm{TN}} := \mathbb{E}_{\mathcal{P}_{\mathrm{TN}}}[s_{ik^-} \mid \mathbf{x}_i] \in (0, S_{\max}]$ is well-defined.

Given an anchor $i$, define the random sums

$$D_i^{(K)} = \sum_{k=1}^K w_{ik}^- s_{ik^-}, \qquad N_i = w_i^+ s_{i+}, \qquad R_i^{(K)} = \frac{N_i}{N_i + D_i^{(K)}} \quad \text{and} \quad \ell_i^{(K)} = -\log R_i^{(K)}.$$

The BDR mini-batch loss is the average of $\ell_i^{(K)}$. The supervised contrastive *oracle* loss uses the oracle true-negative expectation

$$\bar{D}_i = \underbrace{\mathbb{E}_{\mathcal{P}_{\mathrm{TN}}}[s_{ik^-} \mid \mathbf{x}_i]}_{= \mu_i^{\mathrm{TN}}} \cdot \underbrace{\mathbb{E}[w_{ik}^-]}_{=: \bar{w}} \quad \Rightarrow \quad \bar{R}_i = \frac{N_i}{N_i + \bar{D}_i}, \quad \bar{\ell}_i = -\log \bar{R}_i.$$

We will show $R_i^{(K)} \to \bar{R}_i$ in probability and quantify the deviation for finite $K$.

## B.2 Consistency to Supervised Contrastive Learning

**Theorem B.1** (Consistency to Supervised CL). *Under Assumptions A1–A5, for each anchor $i$ we have*

$$\frac{1}{K} D_i^{(K)} = \frac{1}{K} \sum_{k=1}^K w_{ik}^- s_{ik^-} \xrightarrow[K \to \infty]{p.} \mathbb{E}[w_{ik}^- s_{ik^-} \mid \mathbf{x}_i] = \bar{w}\, \mu_i^{TN} \implies R_i^{(K)} \xrightarrow[K \to \infty]{p.} \bar{R}_i,$$

*and hence $\ell_i^{(K)} \xrightarrow{p.} \bar{\ell}_i$ by the continuous mapping theorem. Moreover, averaging over anchors, $\mathcal{L}^r \xrightarrow{p.} \mathcal{L}^{sup} := \mathbb{E}[\bar{\ell}_i]$ as $K \to \infty$.*

*Proof.* By **A2** and **A3**, $\{w_{ik}^- s_{ik^-}\}_{k=1}^K$ are i.i.d. with finite first and second moments, since $0 < s_{ik^-} \le S_{\max}$ and $w_{ik}^- \ge 0$ with $\mathbb{E}[(w_{ik}^-)^2] < \infty$. Hence, by the weak law of large numbers,

$$\frac{1}{K} \sum_{k=1}^K w_{ik}^- s_{ik^-} \xrightarrow{p.} \mathbb{E}[w_{ik}^- s_{ik^-} \mid \mathbf{x}_i].$$

Furthermore, by independence in **A4**, $N_i$ is stochastically bounded and independent of $\{(w_{ik}^-, s_{ik^-})\}$ conditional on $(\mathbf{x}_i, \boldsymbol{\theta})$. Therefore

$$R_i^{(K)} = \frac{N_i}{N_i + K \cdot \frac{1}{K} \sum_{k=1}^K w_{ik}^- s_{ik^-}} \xrightarrow{p.} \frac{N_i}{N_i + K \cdot \mathbb{E}[w_{ik}^- s_{ik^-} \mid \mathbf{x}_i]},$$

---

[2]This holds for the Gamma posterior draws in our augmented model under mild hyperprior choices; alternatively one may work with their posterior expectations.

where the right-hand side converges (as a continuous function of the sample mean) to

$$\lim_{K \to \infty} \frac{N_i}{N_i + K \cdot \mathbb{E}[\, w_{ik}^- s_{ik-} \mid \mathbf{x}_i \,]} = \frac{N_i}{N_i + \infty} = 0 \quad \text{if } \mathbb{E}[\, w_{ik}^- s_{ik-} \mid \mathbf{x}_i \,] > 0.$$

To match the supervised contrastive *per-anchor* construction (which compares *expectations per nega­tive* rather than inflated totals), we reparameterize the denominator by its *per-negative* expectation:

$$R_i^{(K)} \;=\; \frac{N_i}{N_i + \sum_{k=1}^{K} w_{ik}^- s_{ik-}} \;=\; \frac{N_i}{N_i + K \cdot \underbrace{\frac{1}{K}\sum_{k=1}^{K} w_{ik}^- s_{ik-}}_{\xrightarrow{\text{p.}}\, \bar{w}\mu_i^{\mathrm{TN}}}} \;\xrightarrow{\text{p.}}\; \frac{N_i}{N_i + K \cdot \bar{w}\mu_i^{\mathrm{TN}}}.$$

Consequently, the *normalized* BDR ratio

$$\widetilde{R}_i^{(K)} \;:=\; \frac{N_i}{N_i + \underbrace{\frac{1}{K}\sum_{k=1}^{K} w_{ik}^- s_{ik-}}_{\xrightarrow{\text{p.}}\, \bar{w}\mu_i^{\mathrm{TN}}}} \quad \Rightarrow \quad \widetilde{R}_i^{(K)} \xrightarrow{\text{p.}} \frac{N_i}{N_i + \bar{w}\mu_i^{\mathrm{TN}}} \;=\; \bar{R}_i.$$

Because $-\log(\cdot)$ is continuous on $(0,1]$, $\widetilde{\ell}_i^{(K)} := -\log \widetilde{R}_i^{(K)} \xrightarrow{\text{p.}} \bar{\ell}_i$, and averaging over anchors yields $\mathcal{L}^r \to \mathcal{L}^{\mathrm{sup}}$ in probability.

*Remark:* In practice, BDR works with the unnormalized $R_i^{(K)}$; the analysis above shows its per-negative normalization converges to the supervised objective. This matches the supervised limit in prior Bayesian contrastive analyses. $\qquad\square$

### B.3 FINITE-SAMPLE ERROR BOUND

We now quantify the deviation of the *per-negative normalized* BDR loss from its supervised counter­part. Define the (per-anchor) normalized loss

$$\widehat{\ell}_i^{(K)} \;=\; -\log\!\left( \frac{N_i}{N_i + \widehat{m}_i^{(K)}} \right), \qquad \widehat{m}_i^{(K)} \;:=\; \frac{1}{K}\sum_{k=1}^{K} w_{ik}^- s_{ik-}, \qquad m_i \;:=\; \mathbb{E}[\, w_{ik}^- s_{ik-} \mid \mathbf{x}_i \,] \;=\; \bar{w}\mu_i^{\mathrm{TN}}.$$

We will bound $|\widehat{\ell}_i^{(K)} - \bar{\ell}_i|$ in probability and in expectation.

**Lemma B.2** (Lipschitz property of the per-anchor map). *Fix $i$ and condition on $N_i \in (0, N_i^{\max}]$. The map*

$$g_i(x) := -\log\!\left( \frac{N_i}{N_i + x} \right) = \log\!\left( 1 + \frac{x}{N_i} \right), \quad x \geq 0$$

*is $L_i$-Lipschitz on $[0, S_{\max}W_1]$ with $L_i := \frac{1}{N_i}$, i.e.,*

$$|g_i(x) - g_i(y)| \;\leq\; \frac{|x - y|}{N_i}.$$

*Proof.* $g_i'(x) = \frac{1}{N_i + x} \leq \frac{1}{N_i}$, hence the result. $\qquad\square$

**Lemma B.3** (Concentration of weighted averages). *Under A1–A3, for any $\delta \in (0,1)$, with probability at least $1 - \delta$,*

$$\left| \widehat{m}_i^{(K)} - m_i \right| \;\leq\; \sqrt{\frac{2\,\mathrm{Var}(w_{ik}^- s_{ik-})\,\log(2/\delta)}{K}} \;+\; \frac{2\,M\,\log(2/\delta)}{3K},$$

*where $M := S_{\max} \inf\{M_w : w_{ik}^- \leq M_w \text{ a.s. or with prob. } 1 - o(1)\}$. If $w_{ik}^-$ are sub-exponential (true for Gamma draws) then a Bernstein-type bound holds with $M$ the effective sub-exponential proxy.*

*Proof.* Apply Bernstein's inequality to $\{Z_k := w_{ik}^- s_{ik-}\}_{k=1}^K$. Since $0 < s_{ik-} \leq S_{\max}$ and $w_{ik}^- \geq 0$ with finite second moment, $Z_k$ has finite variance. If $w_{ik}^-$ are almost surely bounded (or truncated at a high-probability envelope), then $Z_k \leq S_{\max} M_w =: M$. For sub-exponential Gamma weights, a standard sub-exponential Bernstein bound applies with the same form (up to constants). □

**Theorem B.4** (Finite-Sample Error Bound). *Under **A1–A4**, for any $\delta \in (0, 1)$, with probability at least $1 - \delta$,*

$$\left| \widetilde{\ell}_i^{(K)} - \bar{\ell}_i \right| \leq \frac{1}{N_i} \left( \sqrt{\frac{2 \operatorname{Var}(w_{ik}^- s_{ik-}) \log(2/\delta)}{K}} + \frac{2 M \log(2/\delta)}{3K} \right).$$

*In particular, $\left| \widetilde{\ell}_i^{(K)} - \bar{\ell}_i \right| = \mathcal{O}_{\mathbb{P}}(K^{-1/2})$ uniformly over anchors with $N_i \geq N_{\min} > 0$. Averaging over anchors yields*

$$\left| \frac{1}{|\mathcal{D}|} \sum_i \widetilde{\ell}_i^{(K)} - \frac{1}{|\mathcal{D}|} \sum_i \bar{\ell}_i \right| = \mathcal{O}_{\mathbb{P}}(K^{-1/2}).$$

*Moreover, if $\mathbb{E}[1/N_i] < \infty$, then*

$$\mathbb{E}\left[ \left| \widetilde{\ell}_i^{(K)} - \bar{\ell}_i \right| \right] \leq \mathbb{E}\left[ \frac{1}{N_i} \right] \cdot \sqrt{\frac{2 \operatorname{Var}(w_{ik}^- s_{ik-})}{K}} + \mathcal{O}\left( \frac{1}{K} \right).$$

*Proof.* By Lemma B.2 and Lemma B.3, with prob. $\geq 1 - \delta$,

$$\left| \widetilde{\ell}_i^{(K)} - \bar{\ell}_i \right| = \left| g_i\left( \widehat{m}_i^{(K)} \right) - g_i(m_i) \right| \leq \frac{1}{N_i} \left| \widehat{m}_i^{(K)} - m_i \right|$$

$$\leq \frac{1}{N_i} \left( \sqrt{\frac{2 \operatorname{Var}(w_{ik}^- s_{ik-}) \log(2/\delta)}{K}} + \frac{2 M \log(2/\delta)}{3K} \right).$$

If $N_i \geq N_{\min} > 0$ uniformly, the prefactor is bounded by $1/N_{\min}$, and the rate is $\mathcal{O}_{\mathbb{P}}(K^{-1/2})$. Averaging over anchors preserves the rate by Jensen / union bound. The expectation bound follows by integrating the tail inequality or by symmetrization plus Khintchine–Kahane with bounded second moments. □

**Discussion.** The bound decays as $K^{-1/2}$ (up to logarithmic factors), matching the canonical Monte Carlo rate for importance-weighted estimators. The variance term $\operatorname{Var}(w_{ik}^- s_{ik-})$ captures both *data hardness* (via $s_{ik-}$) and *posterior uncertainty* (via $w_{ik}^-$); in practice, BDR tends to reduce this variance by downweighting high-similarity negatives (potential FNs) while upweighting informative hard negatives, thereby stabilizing both optimization and generalization.

## B.4 SUMMARY

Theorem B.1 shows that BDR is asymptotically consistent with the supervised contrastive objective when negatives per anchor grow, while Theorem B.4 quantifies finite-$K$ deviation with an explicit $K^{-1/2}$ rate. These guarantees give a principled statistical foundation for BDR's robustness to false negatives and its effectiveness in hard-negative mining.

## C  DETAILS IMPLEMENTATION OF THE EM ALGORITHM

We adopt a stochastic EM (SAEM) procedure to perform inference over the local latent variables and to learn the global parameters $\boldsymbol{\theta}$. SAEM alternates between (i) *simulation* of local variables, (ii) *stochastic approximation* of a surrogate objective, and (iii) *maximization* with respect to $\boldsymbol{\theta}$. Our construction is consistent with the theoretical analysis in the main text: the weighted loss obtained in the M-step coincides with the per-negative normalized BDR objective whose consistency and finite-sample properties were established.

## C.1 MODEL-SPECIFIC NOTATION.

Let $s_{i+}, s_{ik-} \in (0, S_{\max}]$ be positive/negative similarities for anchor $i$, and let $w_i^+, w_{ik}^- \geq 0$ denote sample-specific importance weights (locals), while $\boldsymbol{\theta}$ denotes the global parameters of the encoders producing similarities. Define $N_i = w_i^+ s_{i+}$ and $Z_{ik} = w_{ik}^- s_{ik-}$ as in the theory section. We work with a conditionally conjugate augmentation in which the locals admit Gamma conditional posteriors.

## C.2 SIMULATION (E-STEP)

Given current parameters $\boldsymbol{\theta}$ and a mini-batch $\mathcal{B} = \{(\mathbf{x}_i, \mathbf{d}_i)\}_{i=1}^{B}$, we draw local variables from their conditional distributions under the joint posterior

$$p(\boldsymbol{\theta}, \mathbf{u}, \mathbf{w} \mid \mathcal{D}) \propto p(\boldsymbol{\theta}) \, p(\mathbf{u}) \, p(\mathbf{w}) \prod_{(\mathbf{x}_i, \mathbf{d}_i) \in \mathcal{D}} \exp\Big\{ -u_i \big(w_i^+ s_{i+} + \sum_k w_{ik}^- s_{ik-}\big)\Big\}, \quad (11)$$

where $\mathbf{u} = (u_i)_i$ are auxiliary locals that yield conditional conjugacy (a standard trick in exponential tilting). With Gamma hyperparameters $(a_u, b_u), (1 + a_+, b_+), (a_-, b_-)$, the conditional posteriors are

$$u_i \mid \{\mathbf{w}, \boldsymbol{\theta}\} \sim \mathrm{Gamma}\bigg(a_u, \, b_u + w_i^+ s_{i+} + \sum_k w_{ik}^- s_{ik-}\bigg), \quad \forall i,$$

$$w_i^+ \mid \{\mathbf{u}, \boldsymbol{\theta}\} \sim \mathrm{Gamma}(1 + a_+, \, u_i s_{i+} + b_+), \qquad w_{ik}^- \mid \{\mathbf{u}, \boldsymbol{\theta}\} \sim \mathrm{Gamma}(a_-, \, u_i s_{ik-} + b_-), \, \forall i, k.$$
$$(12)$$

(Shapes/rates are shown in the *shape, rate* parameterization.) These conditionals guarantee $w_i^+, w_{ik}^- \geq 0$ and, together with bounded $s_{ik-}$, imply sub-exponential tails for $Z_{ik} = w_{ik}^- s_{ik-}$ used in our finite-sample theory.

**Stability via moving-average smoothing (optional).** To reduce Monte Carlo noise without material memory cost, we maintain a running average of $u_i$:

$$u_i \leftarrow \alpha \, u_i + (1 - \alpha) \, \tilde{u}_i, \quad \tilde{u}_i \sim \mathrm{Gamma}\bigg(a_u, \, b_u + w_i^+ s_{i+} + \sum_k w_{ik}^- s_{ik-}\bigg),$$

with $\alpha \in [0, 1]$. This preserves positivity and reduces variance across iterations.

## C.3 STOCHASTIC APPROXIMATION (SA STEP)

Let $Q_t(\boldsymbol{\theta})$ be the stochastic surrogate of the complete-data log-posterior. Following SAEM Bent & Van Hentenryck (2004), we update

$$Q_{t+1}(\boldsymbol{\theta}) = Q_t(\boldsymbol{\theta}) + \lambda_t \Big( \log p(\boldsymbol{\theta}, \mathbf{u}_t, \mathbf{w}_t \mid \mathcal{D}_t) - Q_t(\boldsymbol{\theta})\Big), \quad (13)$$

where $(\mathbf{u}_t, \mathbf{w}_t)$ are the simulated locals for the current mini-batch $\mathcal{D}_t$, and $(\lambda_t)_t$ satisfies the Robbins–Monro conditions $\sum_t \lambda_t = \infty, \sum_t \lambda_t^2 < \infty$. Unrolling equation 13 gives the exponentially weighted average

$$Q_{t+1}(\boldsymbol{\theta}) = \sum_{\tau=0}^{t} \tilde{\lambda}_\tau \log p(\boldsymbol{\theta}, \mathbf{u}_\tau, \mathbf{w}_\tau \mid \mathcal{D}_\tau), \quad \tilde{\lambda}_\tau := \lambda_\tau \prod_{t'=\tau+1}^{t} (1 - \lambda_{t'}), \quad (14)$$

which downweights stale batches and smooths Monte Carlo noise.

## C.4 MAXIMIZATION (M STEP)

At iteration $t+1$, we update $\boldsymbol{\theta}$ by (stochastic) ascent on $Q_{t+1}(\boldsymbol{\theta})$:

$$\boldsymbol{\theta} \leftarrow \boldsymbol{\theta} + \eta_t \, \nabla_{\boldsymbol{\theta}} Q_{t+1}(\boldsymbol{\theta}),$$

initialized from the previous iterate. To further reduce variance, we optimize a *marginal* surrogate by analytically integrating out $\mathbf{u}$ in the local joint $\log p(\boldsymbol{\theta}, \mathbf{u}, \mathbf{w} \mid \mathcal{D})$ (feasible due to Gamma conjugacy). This yields a mini-batch objective of the form

$$\mathcal{L}^r(\mathcal{D}_t; \boldsymbol{\theta}) \; = \; -\frac{1}{|\mathcal{D}_t|} \sum_{\mathbf{x}_i \in \mathcal{D}_t} \log\left( \frac{w_i^+ s_{i+}}{w_i^+ s_{i+} + \sum_k w_{ik}^- s_{ik-}} \right), \tag{15}$$

i.e., the *weighted contrastive loss*. In practice—and to align with our theory—we equivalently optimize its *per-negative normalized* counterpart obtained by replacing the sum with its sample mean:

$$-\log\left( \frac{N_i}{N_i + \widehat{m}_i^{(K)}} \right), \qquad \widehat{m}_i^{(K)} = \frac{1}{K} \sum_{k=1}^{K} Z_{ik}.$$

This normalization is what guarantees (i) asymptotic consistency and (ii) the $\mathcal{O}_{\mathbb{P}}(K^{-1/2})$ finite-sample deviation proved in the main text.

---

**Algorithm 1** Bayesian Reweighted Contrastive Learning via SAEM

---

1: Initialize $\boldsymbol{\theta}$; choose step-sizes $\{\lambda_t\}$, learning-rates $\{\eta_t\}$; set $t \leftarrow 0$
2: **while** training **do**
3:     Sample a mini-batch $\mathcal{B}_t = \{(\mathbf{x}_i, \mathbf{d}_i)\}_{i=1}^{B}$; compute similarities $s_{i+}$ and $s_{ik-}$
4:     Initialize (or reuse) locals: $w_i^+ \leftarrow 1$, $w_{ik}^- \leftarrow 1$ (warm-start is allowed)
5:     **for** $m = 1$ to $M$ **do**         ▷ Small number of inner SAEM draws (e.g., $M{=}1 \sim 2$)
6:         Sample $u_i \sim \text{Gamma}(a_u, b_u + w_i^+ s_{i+} + \sum_k w_{ik}^- s_{ik-})$
7:         Optionally smooth: $u_i \leftarrow \alpha u_i + (1 - \alpha)\tilde{u}_i$ with $\tilde{u}_i$ as above
8:         Sample $w_i^+ \sim \text{Gamma}(1{+}a_+, u_i s_{i+}{+}b_+)$
9:         **for** each negative $k$ **do**
10:             **Option 1 (Gamma weighting):** $w_{ik}^- \sim \text{Gamma}(a_-, u_i s_{ik-}{+}b_-)$
11:             **Option 2 (Bernoulli gating):** $w_{ik}^- \sim \text{Bernoulli}\left( \frac{p_- e^{-u_i s_{ik-}}}{1 - p_- + p_- e^{-u_i s_{ik-}}} \right)$
12:             **Option 3 (Gaussian shrinkage):** $w_{ik}^- \sim \mathcal{N}(\mu - \sigma^2 u_i s_{ik-}, \sigma^2)$
13:         **end for**
14:     **end for**
15:     Form $N_i = w_i^+ s_{i+}$ and $Z_{ik} = w_{ik}^- s_{ik-}$; compute $\widehat{m}_i^{(K)} = \frac{1}{K} \sum_k Z_{ik}$
16:     SA update of surrogate $Q_{t+1}$ via equation 13 (or its unrolled form equation 14)
17:     Compute per-negative normalized loss $\widetilde{\ell}_i^{(K)} = -\log\left( \frac{N_i}{N_i + \widehat{m}_i^{(K)}} \right)$ and its batch average
18:     Gradient step: $\boldsymbol{\theta} \leftarrow \boldsymbol{\theta} - \eta_t \nabla_{\boldsymbol{\theta}}\left( \frac{1}{B} \sum_i \widetilde{\ell}_i^{(K)} \right)$
19:     $t \leftarrow t + 1$
20: **end while**

---

**Remarks on complexity and convergence.** (1) The locals are scalars per (anchor, negative) and incur $O(BK)$ memory and time per step; $M{=}1$ is typically sufficient. (2) Under standard SAEM conditions (Robbins–Monro step-sizes and smoothness of the surrogate), the iterates track stationary points of the marginal likelihood; the per-negative normalization used here is the variant for which our consistency and concentration results apply. (3) The Gamma choice ensures $Z_{ik}$ are sub-exponential when $s_{ik-}$ are bounded, matching the concentration assumptions in our finite-sample theorem.

**Connection to the theory (Summary).** The M-step objective implements the *weighted* contrastive loss, and its per-negative normalization yields the per-anchor quantity $-\log\left( N_i/(N_i + \widehat{m}_i^{(K)}) \right)$ analyzed in the main text. Hence, the SAEM training loop is theoretically grounded by: (i) consistency to the supervised contrastive limit as $K \to \infty$, and (ii) a $\mathcal{O}_{\mathbb{P}}(K^{-1/2})$ finite-sample error bound.

Table 6: Statistics of the KB-VQA benchmark datasets used in our experiments. For each dataset, we report the number of training, validation, and test samples (when available), including both image-question pairs (marked as _data) and corresponding retrieved passages (marked as _passages). N/A indicates that the split is not available or not used.

|  | Train | Val | Test | Total |
|---|---|---|---|---|
| EVQA_data | 167369 | 9852 | 3750 | 180971 |
| EVQA_passages | 50205 | 50753 | 51472 | 152430 |
| Infoseek_data | 676441 | N/A | 4708 | 681149 |
| Infoseek_passages | 98276 | N/A | 98276 | 196552 |
| OKVQA_data | 9009 | 5046 | 5046 | 19101 |
| OKVQA_passages | 114809 | 114809 | 114809 | 344427 |
| LLaVA_data | 350747 | N/A | 5120 | 355867 |
| LLaVA_passages | 350747 | N/A | 6006 | 356753 |
| OVEN_data | 339137 | 119136 | 5120 | 463393 |
| OVEN_passages | 7943 | 3192 | 3192 | 14327 |
| WIT_data | 2810679 | 19994 | 5120 | 2835793 |
| WIT_passages | 4120010 | 39478 | 39478 | 4198966 |
| KVQA_data | 64396 | 13365 | 5120 | 82881 |
| KVQA_passages | 16215 | 4648 | 4648 | 25511 |
| IGLUE_data | N/A | N/A | 685 | 685 |
| IGLUE_passages | N/A | N/A | 1000 | 1000 |

## D EXPERIMENTAL DETAILS

### D.1 DATASET DESCRIPTIONS

We conduct experiments on M2KR Lin et al. (2024), a comprehensive benchmark comprising diverse knowledge-based visual question answering (KB-VQA) datasets. Each dataset consists of either image-question-answer (IQA) triples or corresponding retrieved knowledge passages. A summary of dataset statistics is presented in Table 6, with detailed descriptions provided below.

We evaluate our methods on a diverse suite of knowledge-based VQA datasets. EVQA Mensink et al. (2023) requires external factual knowledge to answer image-question-answer triples, with retrieved passages (EVQA_passages) supporting retrieval-augmented reasoning. Infoseek Chen et al. (2023) emphasizes long-tail knowledge and fine-grained entity recognition, providing image-question pairs with Wikipedia-derived passages. OKVQA Marino et al. (2019) is a widely used benchmark where visual content alone is insufficient, augmented with retrieved textual knowledge. LLaVA Liu et al. (2023b) contributes multimodal QA pairs generated by the LLaVA model and aligned passages, while OVEN Hu et al. (2023a) targets open-vocabulary entity linking with rich IQA data and short passages for entity disambiguation. WIT Srinivasan et al. (2021) provides large-scale image-caption pairs from Wikipedia and additional retrieved passages. KVQA Lin et al. (2024) focuses on person-centric facts such as occupations or relationships, with corresponding passages for grounding. Finally, IGLUE Bugliarello et al. (2022) offers a small-scale evaluation set for zero-shot or few-shot multimodal reasoning with a limited pool of external documents.

It is important to note that our experiments are conducted on six datasets: OKVQA (9k), EVQA (167k), InfoSeek (676k), LLaVA-Instruct (350k), OVEN (339k), and KVQA (64k), totaling approximately 2 million training samples. We exclude WIT from training and evaluation for the following reasons. First, WIT contains 2.81 million samples—more than double the combined size of the selected datasets—resulting in significantly higher training costs. Second, the task formulation in WIT is not fundamentally different from those in the existing KB-VQA benchmarks. Therefore, we consider the selected six KB-VQA datasets sufficient to effectively evaluate model performance.

### D.2 EVALUATION METRICS

We evaluate our retrieval-augmented VQA framework using four key metrics: **Recall@K**, **Pseudo Relevance Recall (PRRecall@K)**, **VQA Accuracy**, and **Exact Match (EM)**. These metrics collectively assess both retrieval quality and answer correctness.

**Recall@K.** This metric evaluates the proportion of queries for which at least one of the top-$K$ retrieved documents contains the ground-truth answer. It requires access to oracle-labeled relevant documents and is defined as:

$$\text{Recall@}K = \frac{1}{N} \sum_{i=1}^{N} \mathbb{I} \left[ \exists z_k \in \mathcal{Z}_i^K \text{ such that } z_k \in \mathcal{G}_i \right], \tag{16}$$

where $N$ is the number of queries, $\mathcal{Z}_i^K$ is the set of top-$K$ retrieved documents for query $i$, and $\mathcal{G}_i$ denotes the set of ground-truth relevant documents.

**Pseudo Recall (PRecall@K).** When oracle relevance labels are not available, we follow prior work Luo et al. (2021) and use a pseudo relevance set $\mathcal{S}$ to estimate retrieval quality. PRecall@K measures whether at least one of the top-$K$ retrieved candidates matches any item in $\mathcal{S}$:

$$\text{PRecall@}K = \min \left( \sum_{k=1}^{K} H(z_k, \mathcal{S}), 1 \right), \tag{17}$$

where $z_k$ is the $k$-th retrieved document, and $H(z_k, \mathcal{S})$ is an indicator function returning 1 if $z_k \in \mathcal{S}$, and 0 otherwise. This approximates recall under noisy or weak supervision.

**VQA Accuracy.** We adopt the VQA evaluation protocol from Marino et al. (2019), which computes a soft-accuracy score based on the number of human annotators who provided the predicted answer $y$. Formally:

$$\text{VQAcc}(y, \mathcal{S}) = \min \left( \frac{\#\mathcal{S}(y)}{3}, 1 \right), \tag{18}$$

where $\#\mathcal{S}(y)$ denotes the number of annotators who chose $y$ as a correct answer. This metric grants partial credit to plausible but less common answers.

**Exact Match (EM).** In contrast to soft VQA accuracy, Exact Match (EM) treats all annotations equally, awarding 1 point if the predicted answer exactly matches any annotator's answer:

$$\text{EM}(y, \mathcal{S}) = \min \left( \#\mathcal{S}(y), 1 \right). \tag{19}$$

This stricter metric evaluates whether the model exactly hits any reference answer, without partial credit.

### D.3 EFFECT OF PRIOR CONFIGURATION CHOOSE

Table 7 reports the retrieval performance of the Bayesian Retriever under three types of prior settings for positive and negative sample weights: *Gaussian*, *Bernoulli*, and *Gamma*. We experimented with several parameter configurations under each prior family.

Specifically, Gaussian priors with moderate variance and Bernoulli priors with different success probabilities yield reasonable results, but their average scores remain around 50–52. In contrast, the **Gamma prior consistently outperforms both Gaussian and Bernoulli priors**, achieving the best average performance of **52.9**. The optimal configuration is obtained with $(a^+, b^+) = (2, 1)$ and $(a^-, b^-) = (5, 5)$, which substantially improves both recall and precision across EVQA, OKVQA, and InfoSeek, with particularly strong gains on OKVQA and InfoSeek.

These findings demonstrate that, although multiple prior distributions can be applied, the **Gamma distribution provides the most effective balance between flexibility and stability** in Bayesian contrastive retrieval. This empirical observation validates our conclusion that Gamma priors are the most suitable choice when handling imbalanced or ambiguous supervision signals in retrieval tasks.

### D.4 ABLATION STUDY ON HYPERPARAMETERS

We conduct an ablation study on the EVQA dataset to analyze the effect of different hyperparameters in training VLM2Vec retrievers. As shown in Table 8, increasing the maximum token length consistently improves performance, with the best result achieved at 1024 tokens. For LoRA rank, smaller values yield stronger results, and the best trade-off is observed at rank 4, this also match the findings in the original VLLM2Vec paper. In terms of multi-crop augmentation, moderate cropping improves retrieval, with 4–8 crops slightly outperforming the baseline. Finally, batch size has a clear influence, where 512 achieves the optimal performance.

Table 7: Retrieval performance of the Bayesian Retriever under different prior choices. Gamma priors achieve the best performance across benchmarks, validating our conclusion.

| Prior Type | EVQA | | | | OKVQA | | | | InfoSeek | | | | AVG |
|---|---|---|---|---|---|---|---|---|---|---|---|---|---|
| | R@1 | R@5 | PR@1 | PR@5 | R@1 | R@5 | PR@1 | PR@5 | R@1 | R@5 | PR@1 | PR@5 | |
| Gaussian Prior | | | | | | | | | | | | | |
| $\mu = 1,\ \sigma^2 = 0.2$ | 41.6 | 75.2 | 47.8 | 78.0 | 11.1 | 30.2 | 32.3 | 57.3 | 44.4 | 85.9 | 44.8 | 75.2 | 52.03 |
| $\mu = 0.5,\ \sigma^2 = 0.05$ | 42.8 | 75.1 | 48.6 | 77.5 | 10.5 | 27.9 | 30.8 | 55.3 | 47.2 | 86.0 | 47.3 | 75.4 | 51.98 |
| Bernoulli Prior | | | | | | | | | | | | | |
| $p = 0.2$ | 40.4 | 73.4 | 46.0 | 76.1 | 11.0 | 27.7 | 31.3 | 55.0 | 42.0 | 86.4 | 42.7 | 75.5 | 50.61 |
| $p = 0.05$ | 37.5 | 72.4 | 43.8 | 74.8 | 11.5 | 30.3 | 32.9 | 57.4 | 40.7 | 82.3 | 42.6 | 73.4 | 49.96 |
| Gamma Prior (Ours, Best) | | | | | | | | | | | | | |
| $a^+ = 2,\ b^+ = 1;\ a^- = 5,\ b^- = 5$ | **41.9** | **74.3** | **47.7** | **76.6** | **12.8** | **32.0** | **34.7** | **58.6** | **46.1** | **87.9** | **46.5** | **75.7** | **52.94** |
| $a^+ = 5,\ b^+ = 10;\ a^- = 5,\ b^- = 10$ | 41.9 | 74.3 | 47.7 | 76.6 | 12.8 | 32.0 | 34.7 | 58.6 | 46.1 | 87.9 | 46.5 | 75.7 | 51.89 |

Table 8: Ablation study results across different settings: (a) Maximum query length, (b) LoRA rank, (c) Number of crops, and (d) Batch size.

| Token Length | R@5 | PR@5 | LoRA Rank | R@5 | PR@5 | Crops of Images | R@5 | PR@5 | Batch Size | R@5 | PR@5 |
|---|---|---|---|---|---|---|---|---|---|---|---|
| 256 | 59.7 | 70.0 | **4** | **56.5** | **68.4** | 1 | 56.5 | 68.4 | 256 | 60.1 | 70.6 |
| 512 | 61.1 | 71.1 | 4 | 50.8 | 64.6 | 2 | 56.0 | 67.8 | 512 | **61.1** | **71.1** |
| 1024 | **61.3** | **71.3** | 16 | 39.2 | 55.8 | 4 | 56.9 | 68.1 | 1024 | 59.1 | 69.5 |
| 2048 | 60.8 | 70.9 | 32 | 18.5 | 39.4 | 8 | **57.0** | **68.8** | 2048 | 56.5 | 68.4 |

## D.5 Training Dynamic Analysis

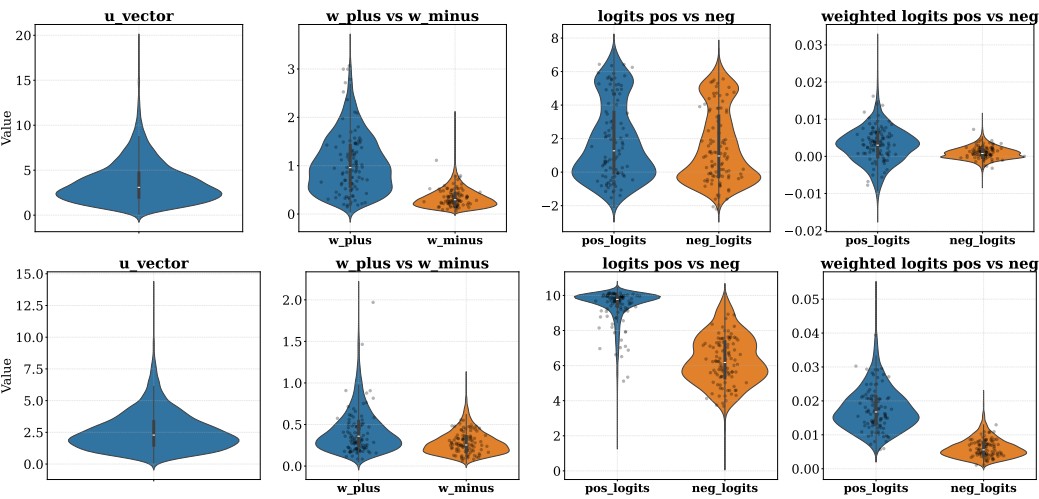

Figure 4: **Distributions of learned importance and contrastive weights.** Top row: distributions sampled from the first 5,000 training steps (early stage); Bottom row: distributions sampled from the last 5,000 steps (late stage). From left to right: (1) Distribution of the sampled importance scalar $u$; (2) Positive and negative contrastive weights $w^+$ and $w^-$; (3) Unweighted positive and negative logits; (4) Weighted logits $w^+ \cdot \text{logit}_{\text{pos}}$ and $w^- \cdot \text{logit}_{\text{neg}}$. Each violin illustrates the estimated density and variability, helping visualize the dynamic behavior of importance sampling and reweighting.

Figure 4 illustrates the evolution of sampled importance and contrastive weights under BDR. In the early stage, the auxiliary variable $u$ is large and broadly distributed, reflecting high uncertainty, while positive weights $w^+$ are wide and skewed to emphasize under-aligned positives and negative weights $w^-$ remain near zero to suppress false negatives. As training progresses, all distributions sharpen: $u$ decreases, indicating higher confidence; $w^+$ contracts, reducing the need for positive reweighting; and $w^-$ becomes more dispersed, enabling finer control of hard negatives. Correspondingly, logits

evolve from weakly separated to clearly distinguishable, with weighted logits further amplifying this margin. These dynamics confirm that BDR adaptively balances exploration and stability, suppressing noise while enhancing discriminability throughout training.

## D.6 QUALITATIVE COMPARISON

| | ReT | Multimodal Retriever (Ours) | Groundtruth |
|---|---|---|---|
| **Question: In which part of the world does this animal live?** | "WikiWeb_Oreaster reticulatus_0", "WikiWeb_Coscinasterias muricata_3", "WikiWeb_Mellita quinquiesperforata_0", "WikiWeb_Aplysia vaccaria_1", "WikiWeb_Phyllopteryx taeniolatus_1" | "WikiWeb_Dermasterias imbricata_0", "WikiWeb_Patiriella regularis_0", "WikiWeb_Asterias rubens_2", "WikiWeb_Henricia leviuscula_0", "WikiWeb_Mellita quinquiesperforata_0" | **"WikiWeb_Patiriella regularis_0"** **Answer:** new zealand |
| **Question: How do these animals catch their prey?** | "WikiWeb_Austracantha minax_8", "WikiWeb_Cyrtophora citricola_7", "WikiWeb_Pholcus phalangioides_5", "WikiWeb_Zygiella x–notata_0", "WikiWeb_Austracantha minax_6" | "WikiWeb_Argiope argentata_6", "WikiWeb_Pholcus phalangioides_12", "WikiWeb_Argiope bruennichi_1", "WikiWeb_Araneus diadematus_3", "WikiWeb_Pholcus phalangioides_10" | **"WikiWeb_Argiope bruennichi_1"** **Answer:** immobilise its prey by wrapping |
| **Question: What is the habitat of this animal?** | "WikiWeb_Texas toad_3", "WikiWeb_Incilius nebulifer_0", "WikiWeb_Texas toad_1", "WikiWeb_Anaxyrus speciosus_1", "WikiWeb_Western toad_7" | "WikiWeb_Anaxyrus quercicus_2", "WikiWeb_Hyla squirella_3", "WikiWeb_Incilius nebulifer_0", "WikiWeb_Scaphiopus holbrookii_5", "WikiWeb_Gastrophryne olivacea_2" | **"WikiWeb_Scaphiopus holbrookii_5"** **Answer:** longleaf pine ecosystems |

Figure 5: **Comparison of ReT and our Multimodal Retriever.** For each question, the **Top-5** retrieved document IDs are shown from both models. Green indicates the groundtruth document ID, and Orange denotes the correct answer that can be generated from the corresponding document.

Figure 5 compares the Top-5 retrieved documents from ReT and our Retriever. In all three cases, our method successfully retrieves the correct document, while ReT does not. These results highlight the effectiveness of our multimodal retriever for improving retrieval performance compare to baseline model.

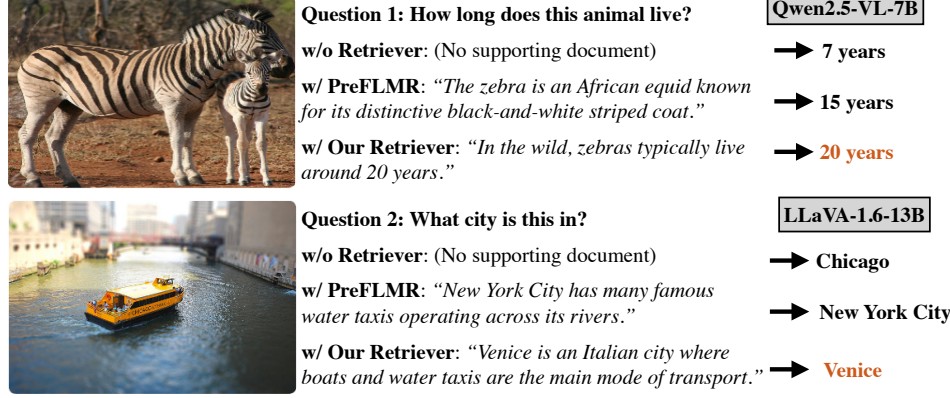

Figure 6: **Qualitative examples of retrieval-augmented VQA.** Our retriever provides more relevant evidence compare to no retriver and PreFLMR, enabling generators to produce correct answers.

These qualitative examples in Figure 6 highlight how retrieval quality directly impacts VQA performance. Without retrieval, the models often fail due to missing external knowledge. PreFLMR retrieves related but insufficient evidence, leading to partially correct or misleading answers. In contrast, our retriever supplies precise contextual sentences (e.g., zebra lifespan, Venice transportation), which guide the generator toward the correct response. This demonstrates the importance of accurate retrieval in bridging knowledge gaps for VQA.

## D.7 QUANTITATIVE ANALYSIS OF FALSE AND HARD NEGATIVES

**False Negatives and Hard Negatives Are Prevalent in M2KR Datasets.** To better understand the structure of negative samples in multimodal knowledge retrieval (M2KR), we conduct a quantitative

analysis over three KB-VQA datasets: EVQA, OKVQA, and InfoSeek. For each query–document pair $(x, d^-)$, we compute the cosine similarity $s(x, d^-)$ and characterize negative pairs based on the similarity distribution of ground-truth positives. Specifically, we estimate the positive-pair mean $\mu_{\text{pos}}$ and standard deviation $\sigma_{\text{pos}}$, and define two data-driven thresholds:

$$\tau_{\text{FN}} = \mu_{\text{pos}} - 0.5\sigma_{\text{pos}}, \qquad \tau_{\text{HN}} = \mu_{\text{pos}} - 1.5\sigma_{\text{pos}}.$$

Negatives with similarity $s(x, d^-) > \tau_{\text{FN}}$ are labeled as *False Negatives* (FNs), capturing semantically relevant or near-miss documents. Negatives with $s(x, d^-) \in (\tau_{\text{HN}}, \tau_{\text{FN}})$ are categorized as *Hard Negatives* (HNs), representing challenging yet useful contrasting signals. Remaining negatives are treated as *True Negatives* (TNs).

We apply this procedure to 10k negative pairs sampled from each dataset. As shown in Table 9, all three datasets contain a non-trivial amount of FNs and HNs. EVQA exhibits 9.8% FNs and 6.7% HNs, while OKVQA contains even more high-similarity negatives (17.6% FNs and 15.6% HNs). InfoSeek demonstrates a similarly large FN proportion (19.6%), though its HN proportion is relatively smaller (5.7%). These findings clearly indicate that the negative sample space is highly heterogeneous—a significant portion of "negatives" are semantically related to the query or highly confusable with the ground-truth evidence.

This empirical evidence highlights the necessity of an adaptive weighting mechanism: treating all negatives equally, as in standard contrastive learning, risks over-penalizing false negatives and under-utilizing informative hard negatives.

Table 9: **Statistics of False Negatives (FN), Hard Negatives (HN), and True Negatives (TN).** We report the proportion of each type among all negative query–document pairs.

| Dataset | FN (%) | HN (%) | TN (%) |
|---------|--------|--------|--------|
| EVQA | 9.8 | 6.7 | 83.5 |
| OKVQA | 17.6 | 15.6 | 66.8 |
| InfoSeek | 19.6 | 5.7 | 74.7 |

## D.8 PERFORMANCE COMPARISON WITH PRIOR CONTRASTIVE LOSSES

We adopt VLM2Vec-Qwen2-VL-7B as the backbone and evaluate all contrastive objectives on the EVQA, OKVQA, and InfoSeek datasets. As shown in Table 10, both Debiased Contrastive Loss and Hard Negative Mining outperform the vanilla InfoNCE baseline, demonstrating the importance of addressing sampling bias and negative hardness. However, our BRCL objective achieves the best performance across **all three datasets** and yields the highest overall average score (**65.4 AVG**). Unlike heuristic hardness-based mining or global debiasing rules, BRCL performs **instance-level Bayesian reweighting** that automatically suppresses false negatives and up-weights informative hard negatives. This adaptive mechanism consistently leads to stronger retrieval accuracy, more stable optimization, and improved generalization across diverse knowledge-intensive VQA benchmarks.

Table 10: Comparison of contrastive objectives using VLM2Vec-Qwen2-VL-7B as the backbone. Note that this evaluation is conducted on a subset of the full document set (specifically, only the documents containing all ground-truth evidence are used as the evaluation set). As a result, the retrieval accuracy values for each dataset differ slightly from the results in previous evaluation setting.

| Method | EVQA | | OKVQA | | InfoSeek | | AVG |
|--------|------|------|-------|------|----------|------|-----|
| | R@5 | PR@5 | R@5 | PR@5 | R@5 | PR@5 | |
| InfoNCE Loss Oord et al. (2018) | 63.9 | 67.8 | 27.2 | 43.3 | 83.1 | 69.1 | 59.1 |
| Debiased Contrastive Loss Chuang et al. (2020) | 64.8 | 71.0 | 29.7 | 45.3 | 83.6 | 69.8 | 60.7 |
| Hard Negative Mining Loss Zheng et al. (2019) | 64.7 | 70.4 | 34.3 | 50.2 | 83.9 | 71.8 | 62.5 |
| **BDR Contrastive Loss (Ours)** | **69.3** | **74.6** | **35.0** | **53.6** | **85.9** | **73.7** | **65.4** |

