# OpenReview forum: "Bayesian Data Reweighting Improves Retrieval in Knowledge-Based VQA"
_ICLR.cc/2026/Conference — Submitted to ICLR 2026_

### Official Review · Reviewer_SpeN · 2025-10-26

**Soundness:** 4
**Presentation:** 4
**Contribution:** 3
**Rating:** 8
**Confidence:** 4

**Summary:**

This paper proposes Bayesian Data Reweighting (BDR) — a probabilistic framework for improving multimodal retrievers in knowledge-based visual question answering (KB-VQA). Instead of treating all non-positive samples equally (as in standard contrastive learning), BDR introduces learnable importance weights for each query-document pair and performs Bayesian inference to adaptively emphasize informative negatives and down-weight false ones.

The authors derive closed-form posterior updates under conjugate priors using an auxiliary variable augmentation scheme, enabling efficient inference through a stochastic Expectation-Maximization (EM) algorithm. They prove theoretical guarantees for asymptotic consistency and finite-sample convergence, and empirically demonstrate that BDR improves retrieval performance across both CLIP-based retrievers (PreFLMR) and LLM-based retrievers (VLM2Vec).

Experiments on multiple KB-VQA datasets (e.g., OKVQA, InfoSeek, EVQA) show consistent gains. Integrating BDR into retrieval-augmented generation also improves downstream VQA accuracy and BLEU scores, sometimes surpassing even large LLMs like GPT-4V in end-task performance.

**Strengths:**

1. BDR introduces a Bayesian probabilistic treatment of sample weighting within contrastive learning, which is an elegant and theoretically grounded approach to mitigate false and hard negatives. The authors provide formal results (conditional conjugacy, consistency, and finite-sample bounds), giving BDR mathematical credibility beyond empirical heuristics.
2. The stochastic EM algorithm allows practical application to large-scale datasets, overcoming scalability concerns typical of Bayesian inference.
3. Evaluation spans multiple retrievers, architectures (CLIP, LLM-based), and datasets, showing consistent and substantial improvements. The study also assesses efficiency, retrieval quality, and VQA accuracy, providing a full picture of impact. BDR improves both retrieval and downstream VQA generation, outperforming prior state-of-the-art retrievers and even some much larger LLM systems.
4. The presentation is good and clear.

**Weaknesses:**

1. Although the stochastic EM is efficient, the paper does not fully quantify its additional training-time cost relative to vanilla contrastive learning or other reweighting baselines.
2. Sections 3.2–4 are mathematically heavy; the presentation could be streamlined. Should be improved to avoid overwhelming readers unfamiliar with Bayesian inference.
3. The study lacks direct comparison with other hard-negative mining or debiased contrastive learning methods under the same multimodal retrieval setting, which would help isolate BDR’s contribution.

**Questions:**

1. How does the proposed approach's runtime compare with standard InfoNCE training (e.g., training time per epoch or GPU-hours)?
2. How are the importance weights initialized during training? Do they converge to stable distributions, or require annealing / regularization to prevent collapse?

---

> ### Author Response · Authors · 2025-11-21
> **Response to Reviewer-SpeN**
>
> We thank the reviewer for the insightful questions. Below we provide detailed responses.
>
> **We sincerely thank the reviewer for recognizing the strength of our theoretical contributions and the consistent improvements achieved across multiple retrievers and KB-VQA datasets. Below, we provide detailed responses to each concern.**
>
> ---
>
> ## W1. Additional training-time cost of stochastic EM
>
> Thank you for raising this important question. **We have added a training efficiency comparison in Section 5.5 of the revised paper to analyze the training efficiency**.
>
> ### Training Efficiency Comparison
>
> | Model        | Backbone      | Steps | Time (h) |
> |--------------|---------------|-------|----------|
> | **Batch Size = 32** | | | |
> | InfoNCE      | Phi-3.5-V     | 521   | 3.17     |
> | BDR (Ours)   | Phi-3.5-V     | 521   | 3.22     |
> | InfoNCE      | Qwen2-VL-7B   | 521   | 2.01     |
> | BDR (Ours)   | Qwen2-VL-7B   | 521   | 2.05     |
>
> | Model        | Backbone      | Steps | Time (h) |
> |--------------|---------------|-------|----------|
> | **Batch Size = 128** | | | |
> | InfoNCE      | Phi-3.5-V     | 131   | 3.16     |
> | BDR (Ours)   | Phi-3.5-V     | 131   | 3.18     |
> | InfoNCE      | Qwen2-VL-7B   | 131   | 2.00     |
> | BDR (Ours)   | Qwen2-VL-7B   | 131   | 2.02     |
>
>
> BDR introduces only lightweight overhead on top of InfoNCE. The stochastic EM updates cost only **O(BK)** per iteration, which is negligible compared to the encoder’s forward–backward complexity. Empirically, the runtime of BDR only slightly longer than InfoNCE. These results confirm that BDR achieves its performance gains **without sacrificing training efficiency**.
>
> ---
>
> ## W2. Mathematical heaviness in Sections 3.2–4
>
> We appreciate this suggestion. We will improve readability by simplifying notation, adding intuitive explanations, and moving long derivations to the Appendix. We will continue revising and updating the paper to ensure the presentation is clearer and easy to follow.
>
>
> ---
>
> ## W3. Missing comparison to hard-negative mining and debiased contrastive learning
>
> We agree and have conducted the requested comparisons, the results have been included in Appendix D.8:
>
> | Method                      | EVQA R@5 | EVQA PR@5 | OKVQA R@5 | OKVQA PR@5 | InfoSeek R@5 | InfoSeek PR@5 | AVG  |
> |-----------------------------|----------|-----------|-----------|------------|--------------|----------------|------|
> | InfoNCE Loss                | 63.9     | 67.8      | 27.2      | 43.3       | 83.1         | 69.1           | 59.1 |
> | Debiased Contrastive Loss   | 64.8     | 71.0      | 29.7      | 45.3       | 83.6         | 69.8           | 60.7 |
> | Hard Negative Mining Loss   | 64.7     | 70.4      | 34.3      | 50.2       | 83.9         | 71.8           | 62.5 |
> | **BDR (Ours)**              | **69.3** | **74.6**  | **35.0**  | **53.6**   | **85.9**     | **73.7**       | **65.4** |
>
> Although both Debiased Contrastive Loss and Hard Negative Mining strategies provide meaningful improvements over standard InfoNCE, our BDR framework achieves superior retrieval performance by performing principled, instance-specific reweighting of negative samples through Bayesian inference.
>
> ---
>
> ## Q2. Initialization and convergence of importance weights
>
> All importance weights in BDR are sampled from conjugate Bayesian priors and therefore require **no manual initialization**. During training, the weights converge stably—false negatives are suppressed, hard negatives retain moderate values, and auxiliary variables $u_i$ quickly stabilize. Because the posterior updates are inherently self-regularizing, **no annealing or additional regularization is needed** for any dataset or backbone.
>
> We thank the reviewer again for the thoughtful and constructive feedback.

---

> > ### Comment · Reviewer_SpeN · 2025-11-23
> >
> > Thanks for the response. Please make sure all the promised changes will take effect. I will keep my score.

---

### Official Review · Reviewer_ygZC · 2025-10-31

**Soundness:** 2
**Presentation:** 2
**Contribution:** 3
**Rating:** 4
**Confidence:** 3

**Summary:**

This paper introduces Bayesian Data Reweighting (BDR), a novel framework designed to improve multimodal retrievers for knowledge-based visual question answering (VQA). BDR addresses the limitations of standard contrastive learning by assigning learnable, adaptive importance weights to positive and negative samples through a principled Bayesian inference procedure.

**Strengths:**

- The method demonstrates performance gains across multiple model architectures (CLIP-based and LLM-based)
- Theoretically, it provides proofs for inference via conjugate priors and establishes the statistical consistency of its objective.

**Weaknesses:**

- The empirical improvements attributed to BDR appear marginal in several key scenarios. For instance, on the EVQA dataset in Table 1, the gains over the InfoNCE baseline are minimal. Furthermore, the average improvement reported in Table 3 is modest. These results raise questions about the practical significance and consistent advantage of BDR over strong baselines.
- The experimental evaluation primarily uses the standard InfoNCE loss as the baseline. However, to properly situate BDR's contribution, it is crucial to compare against more advanced methods that also address false and hard negatives, such as the debiased contrastive loss and hardness-aware weighting schemes, which are discussed in the related work. Without such comparisons, the relative merit of the proposed Bayesian approach remains unclear.
- The theoretical analysis relies on the assumption that negative samples are i.i.d. This is a strong and often unrealistic assumption in contrastive learning, where negatives are typically sampled from a shared batch, introducing structured correlations.
- The paper positions BDR as a general framework for contrastive learning. However, its effectiveness is demonstrated solely within the domain of knowledge-based VQA. Evaluation on other canonical contrastive learning tasks may be helpful to substantiate the claim of generality.

**Questions:**

The theoretical analysis proves global properties like consistency but does not formally link the Bayesian weighting mechanism to the core concepts of false and hard negatives. Could the authors provide further insight, either theoretically or empirically, into how the inferred weights correlate with the semantic "hardness" or "falseness" of negatives? A deeper analysis showing that the framework reliably suppresses false negatives (assigning near-zero weights) and up-weights informative hard negatives would significantly strengthen the causal claim behind its success.

---

> ### Author Response · Authors · 2025-11-21
> **Rebuttal to Reviewer-ygZC**
>
> # Rebuttal to Reviewer-ygZC
>
> We appreciate the reviewer for recognizing the theoretical contributions of our Bayesian formulation and the consistent performance gains across different retrieval architectures. Below, we provide detailed responses to each concern.
>
> ## W1. “BDR improvements appear marginal.”
>
> In the revised paper, we have re-organized **Table 1** to clearly show the improvements over the baseline InfoNCE method. While **EVQA** exhibits relatively small gains for the CLIP-based retriever (0.1–0.3), other datasets show much larger improvements—**OKVQA (2.3–4.0)**, **InfoSeek (0.6–6.8)**, and **OVEN (3.3–6.0)**. This is due to EVQA’s dataset characteristics: its passages are highly specific with very low semantic overlap, as shown in Appendix Table 9，which may contains the **fewest false negatives**, leaving limited room for BDR to correct hard or false negatives.
>
> Importantly, BDR delivers **consistent gains** on other datasets and on the **six-dataset average**, surpassing prior state-of-the-art retrievers such as **ReT (58.9)** and **PreFLMR (56.4)**. Moreover, BDR also improves downstream VQA accuracy by **2-3 points** (Table 3), and by **3.5–6.2 points** on InfoSeek and EVQA (Table 2), confirming that enhancing retrieval quality leads to better VQA performance.
>
> Overall, these results highlight that the empirical strength of BDR lies not in isolated improvements on a single dataset, but in its **consistency improvement and robustness** across multiple retrieval and answer generation evaluation settings.
>
> ---
>
> ## W2. Comparison With Other Hardness- and Debiasing-Based Methods
> We appreciate the reviewer highlighting this point. We have now conducted additional experiments on comparing BDR against debiased contrastive loss and Hard Negative Mining Loss, the results are shown below:
>
>
> | Method                      | EVQA R@5 | EVQA PR@5 | OKVQA R@5 | OKVQA PR@5 | InfoSeek R@5 | InfoSeek PR@5 | AVG  |
> |-----------------------------|----------|-----------|-----------|------------|--------------|----------------|------|
> | InfoNCE Loss                | 63.9     | 67.8      | 27.2      | 43.3       | 83.1         | 69.1           | 59.1 |
> | Debiased Contrastive Loss   | 64.8     | 71.0      | 29.7      | 45.3       | 83.6         | 69.8           | 60.7 |
> | Hard Negative Mining Loss   | 64.7     | 70.4      | 34.3      | 50.2       | 83.9         | 71.8           | 62.5 |
> | **BDR (Ours)**              | **69.3** | **74.6**  | **35.0**  | **53.6**   | **85.9**     | **73.7**       | **65.4** |
>
>
> These results show that BDR achieves the **best performance across all three datasets** and obtains the highest overall average score (**65.4 AVG**).
>
> We have included these results in the Appendix Table 10.
>
> ---
>
> ## W3. “i.i.d. negative assumption unrealistic.”
>
> We thank the reviewer for raising this point. While negatives in contrastive learning are sampled from the same mini-batch and thus not strictly independent, our theoretical analysis adopts the standard i.i.d. assumption **only as a technical simplification to enable likelihood factorization and consistency analysis**. Importantly, BDR does not require i.i.d. negatives in practice—the augmented likelihood and conjugate posterior updates remain valid under any mini-batch sampling scheme.
>
> ---
>
> ## W4. “BDR claims to be a general contrastive-learning framework.”
>
> We would like to thank the reviewer for the thoughtful comment.  We would like to clarify that our primary claim is that **Bayesian Data Reweighting improves retrieval performance in knowledge-based VQA**, where false negatives and hard negatives are particularly common in this task. We do not position our method as a general contrastive-learning framework. We agree that extending our approach to broader contrastive-learning settings is valuable, but it requires additional empirical studies and is beyond the current scope. We appreciate the reviewer for pointing out this promising direction for future work.
>
>
>
> ---
>
> ## Q1. “How do weights correlate with false vs. hard negatives?”
>
> We appreciate the reviewer’s question about the connection between Bayesian weights and the semantic hardness or falseness of negatives. To clarify, we conducted a correlation analysis between the inferred negative weights and their semantic attributes. Using the last 100 training steps on OKVQA, the results show clear and meaningful correlations:
>
> - **Hard negatives:** `corr(w^-, HN_probability) = 0.67`, indicating that BDR up-weights informative hard negatives.
>
> - **False negatives:** `corr(w^-, FN_probability) = -0.71`, demonstrating that BDR effectively down-weights semantically relevant false negatives.
>
>
> These results clearly show that BDR suppresses false negatives and up-weights hard negatives, directly explaining its performance gains. This behavior is also reflected in the training dynamics, where BDR enlarges the positive–negative decision margin (Figure 2 and Figure 4 in the paper).

---

### Official Review · Reviewer_s1bH · 2025-11-01

**Soundness:** 2
**Presentation:** 2
**Contribution:** 2
**Rating:** 4
**Confidence:** 4

**Summary:**

This paper focuses on the Knowledge-based Visual Question Answering (KB-VQA) task. The main motivation of this paper is that standard contrastive learning of KB-VQA ignores the potential hierarchical structure of negative pairs including true, false, and hard negatives. These negative pairs need to be handled carefully in the training phase of contrastive learning, therefore a Bayesian reweighting method is proposed by the authors.

**Strengths:**

-	The topic of this paper looks reasonable.
-	The paper is well-written.
-	The experimental results are promising.

**Weaknesses:**

-	The main motivation of this paper is that standard contrastive learning of KB-VQA ignores the potential hierarchical structure of negative pairs including true, false, and hard negatives. Although the authors cite related paper to justify this argument, it lacks obvious quantitative (e.g., the statistics of these hierarchical negative pairs in datasets) or visualized feature embeddings among different pairs to demonstrate the reasonability of this motivation in the introduction or experiment section.
-	For the Efficient Inference with Stochastic Expectation Maximization, it is necessary to formulate the specific complexity using stochastic Expectation-Maximization
(EM) algorithm.
-	For the Augmented Likelihood and Conditional Conjugacy, it is confused that how the data- augmentation is introduced. To me, the random variable $\mu$ is the alternative of explicit sample reweighting and how the random variable links the data-augmentation method? It is a parameter-augmentation method rather than the data-augmentation one? This section is quite mess and it is hard to follow easily.

**Questions:**

NA

---

> ### Author Response · Authors · 2025-11-21
> **Response to Reviewer-s1bH**
>
> # Response to Reviewer-s1bH
>
> We appreciate the reviewer for acknowledging the clarity of the writing, and the promising experimental results. Below we provide our responses to each concern.
>
> ---
>
> ## W1. Motivation & Evidence of Hierarchical Negative Structures
>
> To strengthen the motivation of our work, we conducted a quantitative analysis of **False Negatives (FNs)** and **Hard Negatives (HNs)** in the M2KR datasets, the experiment has been add to Table9 in Appendix~D.7 of the the revised paper. For each query–document pair $(x, d^-)$, we compute the cosine similarity $s(x, d^-)$. Using the similarity distribution of ground-truth positives, we derive data-driven thresholds:
>
> - $\tau_{\text{FN}} = \mu_{\text{pos}} - 0.5\sigma_{\text{pos}}$
> - $\tau_{\text{HN}} = \mu_{\text{pos}} - 1.5\sigma_{\text{pos}}$
>
> This categorizes negatives as:
>
> - **False Negatives (FN):** $s(x, d^-) > \tau_{\text{FN}}$
> - **Hard Negatives (HN):** $\tau_{\text{HN}} < s(x, d^-) < \tau_{\text{FN}}$
> - **True Negatives (TN):** $s(x, d^-) < \tau_{\text{HN}}$
>
> Across 10k query–document pairs in EVQA:
> - FNs: **9.8%**
> - HNs: **6.7%**
> - TNs: **83.5%**
>
> The False negative and hard negative samples also exist in the OKVQA and InfoSeek datasets.These results confirm that negative samples are *not* homogeneous—many are semantically relevant or highly confusable—thus justifying the need for adaptive reweighting.
>
> ---
>
> ## W2. Complexity of Stochastic EM
> We thank the reviewer for pointing out this. **We have added a training cost comparison in Section 5.5 of the revised paper to analyze the computing complexity and training efficiency**. The stochastic EM procedure in BDR is computationally lightweight, and the Stochastic EM updates do not affect overall training speed. Each SAEM iteration consists of three parts:
>
> 1. **E-step (sampling latent variables).**
>    For each instance and its K in-batch negatives, we sample the latent variables
>    $u_i$, $w_i^{+}$, and $w_{ik}^{-}$
>    from closed-form posteriors. All operations are element-wise, giving
>    $O(BK)$ time and memory.
>
> 2. **Stochastic-approximation update.**
>    Updating the sufficient statistics via EMA is a simple vector operation with negligible cost:
>    $O(1)$.
>
> 3. **M-step.**
>    This step is identical to standard InfoNCE: one encoder forward–backward pass with a reweighted contrastive loss. Its complexity remains
>    $O(\text{Encoder-Fwd} + \text{Encoder-Bwd})$,
>    which dominates the total cost.
>
> **Overall complexity.**
> The total cost is:
>
> $$
> O(\text{Encoder}) + O(BK)
> $$
>
> Since $O(\text{Encoder})$ is much larger than $O(BK)$, the SAEM overhead is minimal. Memory usage is also $O(BK)$, because SAEM does not store latent variables beyond the current mini-batch. This matches our empirical observation that BDR adds only a small extra cost compared with InfoNCE.
>
>
> ## W3. Clarification on Augmented Likelihood & Conditional Conjugacy
>
> We thank the reviewer for raising this important point. Our method uses **data augmentation in the classical EM/Gibbs sense**: the auxiliary variable $u_i$ is introduced purely as a latent variable so that the weighted contrastive likelihood becomes a complete-data likelihood with a conjugate Gamma form. This follows the standard Bayesian data-augmentation framework [1], where $u_i$  is integrated out and does **not** act as an additional model parameter. This is fundamentally different from parameter-expansion methods such as PX-DA [2]. We will clarify this distinction in the revised paper.
>
>
> [1] Tanner, M. A., & Wong, W. H. (1987). The Calculation of Posterior Distributions by Data Augmentation. Journal of the American Statistical Association, 82(398), 528–540.
>
> [2] Liu, J. S., & Wu, Y. N. (1999). Parameter Expansion for Data Augmentation. Journal of the American Statistical Association, 94(448), 1264–1274.

---

### Official Review · Reviewer_8HZW · 2025-11-01

**Soundness:** 2
**Presentation:** 2
**Contribution:** 3
**Rating:** 4
**Confidence:** 3

**Summary:**

The authors identify that treating all negative pairs as equally informative can lead to false negative bias, making hard negative mining particularly challenging. To address this issue, the paper introduces a novel Bayesian data reweighting approach that calibrates the contributions of positive and negative samples to improve knowledge retrieval for KB-VQA. An efficient EM algorithm is proposed to estimate the optimal weights for both positive and negative examples. The proposed method achieves superior performance compared to previous approaches on the M2KR benchmark.

**Strengths:**

- Proposes a novel Bayesian data reweighting method for KB-VQA retrieval.

- Demonstrates consistent improvements over baseline retrieval setups across various backbones and two architectures.

**Weaknesses:**

- Please use the correct citation style in LaTeX for ICLR.
- In some results (e.g., on OVEN), the baseline scores are higher and should be bolded in Table 1 instead of the proposed method.
- Regarding the VQA performance on E-VQA, it is unusual that the VQA accuracy is much lower than EM, as they should theoretically be on par. Moreover, previous work typically reports the BEM score, which is the standard metric used in the original E-VQA paper. For fair comparison with prior work, the authors should report BEM instead of accuracy for E-VQA. Additionally, the oracle results reported in the original E-VQA paper are substantially higher under the BEM metric. It would be valuable if the authors could reproduce and report the corresponding BEM scores with Oracle for comparison.
- The different configurations of the Gamma prior appear to yield exact identical performance. Could the authors elaborate on how these hyperparameters influence the final results and provide more details behind their selection?
- Similarly, please provide more detailed insights on the three types of priors, beyond the brief intuition mentioned in the paragraph starting at Line 206.

**Questions:**

- Line 424: What does BRCL stand for?

- How does the training cost (computation/time) compare with the InfoNCE baseline?

- Since the method assigns weights to all examples, could these weights be used to rank examples in the corpus for positive/negative selection or data pruning?

---

> ### Author Response · Authors · 2025-11-21
> **Response to Reviewer-8HZW**
>
> # Response to Reviewer-8HZW
>
> We thank the reviewer for recognizing the novelty of our Bayesian data-reweighting approach and acknowledging its consistent improvements. We address the specific concerns below.
>
> ---
>
> ## W1. Citation Style and Table Formatting
>
> We appreciate the reviewer for pointing out the citation style and bold formatting issues in Table 1. We have fixed them in the revised version.
>
> ---
>
> ## W2. Use of BEM Metric on EVQA
>
> Thanks for the suggestion and we have re-evaluated E-VQA using the **BERT Matching (BEM)** metric, and updated the  metric in the revised version of paper.
>
> ### Updated E-VQA Results (LLaVA-1.6-13B Generator)
>
> | Retriever | VQA Acc | BEM |
> |-----------|---------|------|
> | No Retriever | 2.7 | 69.8 |
> | PreFLMR | 8.7 | 74.3 |
> | ReT | 6.5 | 73.2 |
> | **BDR (ours)** | **9.1** | **77.2** |
> | Oracle | 16.1 | 86.7 |
>
> ### Updated E-VQA Results (Qwen2.5-VL-7B Generator)
>
> | Retriever | VQA Acc | BEM |
> |-----------|---------|------|
> | No Retriever | 4.6 | 65.2 |
> | PreFLMR | 11.5 | 68.3 |
> | ReT | 10.8 | 67.9 |
> | **BDR (ours)** | **14.4** | **71.2** |
> | Oracle | 23.3 | 89.1 |
>
> These results align with the Oracle performance reported in the original E-VQA paper, and BDR consistently improves BEM by ~3 points over existing retrievers, demonstrating that our method enhances downstream VQA accuracy by improving retrieval performance.
>
> ---
>
> ## W3. Effect of Different Gamma Prior Configurations
>
> Thank you for the question. The Gamma–prior settings are not truly identical; the main table rounds numbers for readability. We provide the full-precision results in Appendix Table 7, which show small but non-zero differences. Table 7 is intended to analyze prior hyperparameter sensitivity. We varied the means and variances of the priors and reported the best-performing settings. The final performance varies only slightly because, this is expected as during training, the data likelihood gradually dominates, causing the posterior weights to converge to similar values. This confirms the robustness of our Bayesian formulation.
>
> ---
>
> ## W4. Explanation of the Three Types of Priors
>
> We provide our insights behind the three types of priors:
>
> - **Gamma prior** → supports values in $(0, \infty)$ ideal for modeling *continuous* difficulty levels of negatives and providing flexible weight adjustments.
> - **Bernoulli prior** → outputs \(\{0,1\}\), more aggressive: weight 1 keeps the sample (hard negative), weight 0 masks it (false negative).
> - **Gaussian prior** → This prior reflects the assumption that most false-negative weights lie within a relatively stable interval and approximately follow a symmetrical normal-like distribution. In practice, we use a truncated Gaussian to ensure that the weights remain in the positive domain $(0, \infty)$
> We also include these insights in the revised version for clarity.
>
> ---
>
> ## Q. Clarifications on Questions
>
> ### (Q1)**Line 424: What does BRCL stand for?**
> BRCL refers to the earlier name *Bayesian Reweighted Contrastive Learning*. The final unified name used in the paper is **BDR**.
>
> ### (Q2)**Training Cost Compared to InfoNCE**
> **We have added the training cost comparison in Section 5.5 in the revised paper to analyze Training Efficiency**. Theoretically, BDR introduces only **lightweight overhead** on top of InfoNCE because the Stochastic EM step adds merely an additional **\(O(BK)\)** computation, which is negligible compared with the cost of the CLIP or LLM-based encoders computation.  Empirically, we measured the training time of BDR versus standard InfoNCE under the same training steps and batch size. The overall training time is slightly longer than the InfoNCE baseline.
>
> ### (Q3)**Can the learned weights be used for data selection or pruning?**
> Yes. The posterior weights naturally reflect the informativeness of samples and can be used for data selection. We view this as a promising future direction and will mention it in the discussion section.

---

### Meta-Review · Area_Chair_FxTA · 2026-01-07

**Summary:**

This paper proposes a Bayesian data reweighting framework for Knowledge-based VQA, achieving consistent improvements.
Initially, the paper received scores of 4, 4, 4, and 8 from four reviewers. During the rebuttal phase, three reviewers did not participate in further discussion, and therefore, the final scores remained unchanged.
After carefully examining the manuscript, the review comments, and the authors’ responses, the AC acknowledges that the proposed method has merit within the VQA community. However, the core idea of weighting in contrastive learning is not novel in broader research areas such as representation learning. Moreover, the proposed approach lacks sufficient task-specific design tailored to VQA.
Considering these factors, the AC concludes that the manuscript does not yet meet the acceptance standard of the conference.

**Reviewer Concerns:**

--Addressed--  Reviewer (8HZW, s1bH, SpeN): Training Efficiency

--Outstanding--  Reviewer-- (ygZC): Evaluation on other canonical contrastive learning tasks may be helpful to substantiate the claim of generality.

**Reviewer Scores:**

Regarding the comment that the experimental evaluation relied primarily on the standard InfoNCE loss as the baseline and lacked comparisons with more advanced methods addressing false and hard negatives, the authors provided corresponding experimental results in the rebuttal. These additional experiments directly address the reviewer’s concern. Had the reviewer been able to participate fully in the discussion, this clarification and new evidence would likely have led to a revision of their initial score.

---

### Decision · Program_Chairs · 2026-01-26

Reject